# Generating Less Certain Adversarial Examples Improves Robust Generalization

**Minxing Zhang**                                                          *minxing.zhang@cispa.de*
*CISPA Helmholtz Center for Information Security*

**Michael Backes**                                                         *backes@cispa.de*
*CISPA Helmholtz Center for Information Security*

**Xiao Zhang**                                                             *xiao.zhang@cispa.de*
*CISPA Helmholtz Center for Information Security*

**Reviewed on OpenReview:** *https://openreview.net/forum?id=MMtKOkUML7*

## Abstract

This paper revisits the robust overfitting phenomenon of adversarial training. Observing that models with better robust generalization performance are less certain in predicting adversarially generated training inputs, we argue that overconfidence in predicting adversarial examples is a potential cause. Therefore, we propose a formal definition of adversarial certainty that captures the variance of the model's predicted logits on adversarial examples and hypothesize that generating adversarial examples after the optimization of decreasing adversarial certainty improves robust generalization. Our theoretical analysis of synthetic distributions characterizes the connection between adversarial certainty and robust generalization. Accordingly, built upon the notion of adversarial certainty, we develop a general method to search for models that can generate training-time adversarial inputs with reduced certainty, while maintaining the model's capability in distinguishing adversarial examples. Extensive experiments on image benchmarks demonstrate that our method effectively learns models with consistently improved robustness and mitigates robust overfitting, confirming the importance of generating less certain adversarial examples for robust generalization. Our implementations are available as open-source code at: `https://github.com/TrustMLRG/AdvCertainty`.

## 1 Introduction

Deep neural networks (DNNs) have achieved exceptional performance and have been widely adopted in various applications, including computer vision He et al. (2016), natural language processing Devlin et al. (2019) and recommendation systems Covington et al. (2016). However, DNNs have been shown highly vulnerable to classifying inputs, known as *adversarial examples* Szegedy et al. (2014); Goodfellow et al. (2015), crafted with imperceptible perturbations that are designed to trick the model into making wrong predictions. The prevalence of adversarial examples has raised serious concerns regarding the robustness of DNNs, especially when deployed in security-critical applications such as self-driving cars Chen et al. (2015), biometric facial recognition Komkov & Petiushko (2021) and medical diagnosis Finlayson et al. (2019); Ma et al. (2021). To improve the resilience of deep neural networks against adversarial perturbations, numerous defenses have been proposed, such as distillation Papernot et al. (2016), adversarial detection Ma et al. (2018), feature denoising Xie et al. (2019), randomized smoothing Cohen et al. (2019), and semi-supervised methods Alayrac et al. (2019). Among them, adversarial training Madry et al. (2018); Zhang et al. (2019) is by far the most popular approach to learn robustness against adversarial perturbations. Nevertheless, even the state-of-the-art adversarial training methods Croce et al. (2020); Rebuffi et al. (2021); Wang et al. (2023) cannot achieve satisfactory robustness performance on simple classification tasks like classifying CIFAR-10 images.

Witnessing the empirical challenges for improving model robustness, many recent works focus on understanding the behavior of adversarial training Tu et al. (2019); Gao et al. (2019); Wu et al. (2020); Zhang et al. (2020); Yu et al. (2022). In particular, Rice et al. observed that test robust accuracy of intermediate models produced during adversarial training immediately increases by a large margin after the first learning rate decay but keeps decreasing afterward, known as *robust overfitting* Rice et al. (2020). Robust overfitting has recently attracted a lot of attention, since it is not an issue for standard deep learning but appears to be dominant in adversarial training. Therefore, recognizing the fundamental cause of robust overfitting may provide important insights for designing better ways to produce more robust models. In this paper, we revisit the robust overfitting phenomenon and provide a potential reason for why it happens. More concretely, we observe that models produced during adversarial training tend to be overconfident in predicting the class labels of adversarial inputs, whereas models with better robust generalization exhibit much less significant overconfidence issues. By introducing the notion of *adversarial certainty*, which captures the variation of a model's output logits in predicting adversarial examples generated by the model itself, we provide theoretical evidence and empirical results showing that generating adversarial examples after the optimization of decreasing adversarial certainty helps produce models with improved robust generalization.

**Contributions.** By visualizing the label predictions of adversarial examples generated at different epochs, we observe that adversarial training is prone to produce overconfident models, which further induces decreased test robust accuracy. Thus, we argue that generating training-time adversarial inputs after the optimization of decreasing adversarial certainty can improve robust generalization (Section 3). To study the hypothesis more rigorously, we first introduce a formal definition of adversarial certainty, and then provide theoretical results on synthetic distributions that characterize the connection between adversarial certainty and robust generalization (Section 4).

Built upon the definition of adversarial certainty, we propose a general method to explicitly **D**ecrease **A**dversarial **C**ertainty (DAC) during adversarial training (Section 5). At a high level, DAC is designed to find training-time adversarial examples with lower certainty for improving model robustness. In particular, DAC first finds the steepest descent direction of model weights to decrease adversarial certainty, and then the newly generated adversarial examples with lower certainty are used to optimize model robustness. As the model learns from less certain adversarial examples, the aforementioned overconfidence issue is expected to be largely mitigated. In addition, we provide a correlation analysis between adversarial certainty and robust generalization, which illustrates the importance of imposing proper constraints on model search space for DAC. By conducting extensive experiments on image benchmark datasets, we demonstrate that our method consistently produces more robust models when combined with various adversarial training algorithms, and robust overfitting is significantly mitigated with the involvement of DAC (Section 6.1). Moreover, we find that our proposed adversarial certainty has an implicit effect on existing robustness-enhancing techniques that are even designed based on different insights (Section 6.2). Besides, we provide a more intuitive demonstration of DAC's efficacy (Section 6.3), and update the explicit optimization of adversarial certainty by using a regularization term to improve the efficiency (Section 6.4). These empirical results again indicate the importance of adversarial certainty in understanding adversarial training and bring a further comprehension of our work.

**Notation.** We use lowercase boldfaced letters for vectors, and $\mathbb{1}(\cdot)$ for the indicator function. For any $\boldsymbol{x} \in \mathbb{R}^d$ and $i \in \{1, 2, \ldots, d\}$, let $x_i$ be the $i$-th element of $\boldsymbol{x}$. For any finite-sample set $\mathcal{S}$, let $|\mathcal{S}|$ be the cardinality of $\mathcal{S}$. Let $(\mathcal{X}, \Delta)$ be a metric space, where $\Delta : \mathcal{X} \times \mathcal{X} \to \mathbb{R}$ denotes a distance metric. For any $\boldsymbol{x} \in \mathcal{X}$ and $\epsilon \geq 0$, let $\mathcal{B}_\epsilon(\boldsymbol{x}; \Delta) = \{\boldsymbol{x}' \in \mathcal{X} : \Delta(\boldsymbol{x}', \boldsymbol{x}) \leq \epsilon\}$ be the ball centered at $\boldsymbol{x}$ with radius $\epsilon$ and metric $\Delta$. When $\Delta$ is free of context, we simply write $\mathcal{B}_\epsilon(\boldsymbol{x}) = \mathcal{B}_\epsilon(\boldsymbol{x}; \Delta)$. Let $\mu$ be a probability distribution on $\mathcal{X} \times \mathcal{Y}$, where $\mathcal{Y}$ denotes a label space. The empirical distribution of $\mu$ with respect to a sample set $\mathcal{S}$ is defined as: $\hat{\mu}_\mathcal{S}(\mathcal{C}) = \sum_{(\boldsymbol{x},y) \in \mathcal{S}} \mathbb{1}\big((\boldsymbol{x}, y) \in \mathcal{C}\big)/|\mathcal{S}|$ for any measurable set $\mathcal{C} \subseteq \mathcal{X} \times \mathcal{Y}$. Let $\mathcal{N}(\gamma, \sigma^2)$ be the Gaussian distribution with mean $\gamma$ and standard deviation $\sigma > 0$.

## 2 Related Work

Adversarial training is a promising defense framework for improving model robustness against adversarial examples Goodfellow et al. (2015); Madry et al. (2018); Zhang et al. (2019); Wang et al. (2020); Tramèr et al. (2017); Shafahi et al. (2019); Andriushchenko & Flammarion (2020); Wong et al. (2020); Jin et al. (2022). In

particular, Goodfellow et al. proposed to adversarially train models using perturbations generated by the fast gradient sign method (FGSM) Goodfellow et al. (2015). Later on, Madry et al. incorporated perturbations produced by iterative projected gradient descent (PGD) into adversarial training Madry et al. (2018), which learns models with more reliable and robust performance. Other variants of adversarial training have been proposed, which typically modify the training objective but also use PGD attacks to approximately solve the inner maximization problem. For instance, Zhang et al. designed TRADES, which considers optimizing the standard classification loss while encouraging the decision boundary to be smooth Zhang et al. (2019). Wang et al. proposed MART to emphasize the importance of misclassified examples during adversarial training Wang et al. (2020). In this work, we demonstrate how to improve the robust generalization performance of these adversarial training algorithms by searching for models with lower adversarial certainty.

Apart from improving adversarial training, several recent works focus on understanding robust generalization and leveraging the gained insight to build more robust models Rice et al. (2020); Stutz et al. (2021); Hwang et al. (2021); Chen et al. (2021); Yu et al. (2022); Xu et al. (2023). In particular, Rice et al. discovered that, unlike standard deep learning, robust overfitting is a dominant phenomenon for adversarially-trained DNNs that hinders robust generalization, and advocated the use of early stopping Rice et al. (2020). Wu et al. discovered that the flatness of weight loss landscape is an important factor related to robust generalization, which inspires them to adversarially perturb the model weights during adversarial training Wu et al. (2020). Besides, Tack et al. proposed a consistency regularization term based on data augmentation to mitigate robust overfitting Tack et al. (2022). Our work complements these methods, where we explain why overconfidence in generating adversarial examples is highly related to robust overfitting and illustrate how to improve robust generalization by promoting less certain perturbed inputs for adversarial training. Moreover, we are also aware of two existing works that focus on improving the performance of adversarial training with the consideration of model overconfidence Stutz et al. (2020); Setlur et al. (2022). However, these works target different objectives from ours. More specifically, Stutz et al. developed a confidence-calibrated adversarial training method that achieves better robustness against unseen attacks Stutz et al. (2020). Setlur et al. proposed a regularization technique to maximize the entropy of model predictions on out-of-distribution data with larger perturbations, thus improving model accuracy on unseen examplesSetlur et al. (2022).

## 3 Overconfidence Compromises Robustness

In this section, we first introduce the most relevant concepts, including adversarial robustness, adversarial training and robust overfitting, of which the complete introduction and discussion are detailed in Appendix A. Next, we visualize the label predictions of adversarially trained models by heatmaps, and propose our hypothesis that model overconfidence is a potential cause of the decreased robust generalization in adversarial training, where robust overfitting occurs.

**Preliminaries.** In this work, we focus on the most widely-studied $\ell_p$-norm bounded perturbations, and work with the following definition of *adversarial robustness*:

$$\mathcal{R}_\epsilon(f_\theta; \mu) = 1 - \mathbb{E}_{(\boldsymbol{x}, y) \sim \mu} \max_{\boldsymbol{x}' \in \mathcal{B}_\epsilon(\boldsymbol{x})} \mathbb{1}\big(f_\theta(\boldsymbol{x}') \neq y\big),$$

where $f_\theta$ is an arbitrary classifier, $\mu$ denotes the underlying data distribution, and $\epsilon \geq 0$ captures the adversarial strength. In practice, adversarial robustness estimated based on a set of testing examples $\mathcal{R}_\epsilon(f_\theta; \hat{\mu}_{\mathcal{S}_{te}})$ is typically used as the evaluation metric for measuring the robust generalization of $f_\theta$. *Adversarial training* aims to improve model robustness by training on adversarially-perturbed inputs Goodfellow et al. (2015); Madry et al. (2018); Zhang et al. (2019), which can be formulated as a min-max optimization problem:

$$\min_{\theta \in \Theta} \frac{1}{|\mathcal{S}_{tr}|} \sum_{(\boldsymbol{x}, y) \in \mathcal{S}_{tr}} \max_{\boldsymbol{x}' \in \mathcal{B}_\epsilon(\boldsymbol{x})} L\big(f_\theta, \boldsymbol{x}', y\big), \tag{1}$$

where $\Theta$ represents the model class, $\mathcal{S}_{tr}$ is a set of training examples independently and identically sampled from $\mu$, and $L$ denotes some convex surrogate loss such as cross-entropy. Note that PGD attacks Madry et al. (2018) are typically employed in adversarial training to provide approximated solutions to the inner maximization problem in Equation (1). Nevertheless, PGD-based adversarial training and its variants Madry

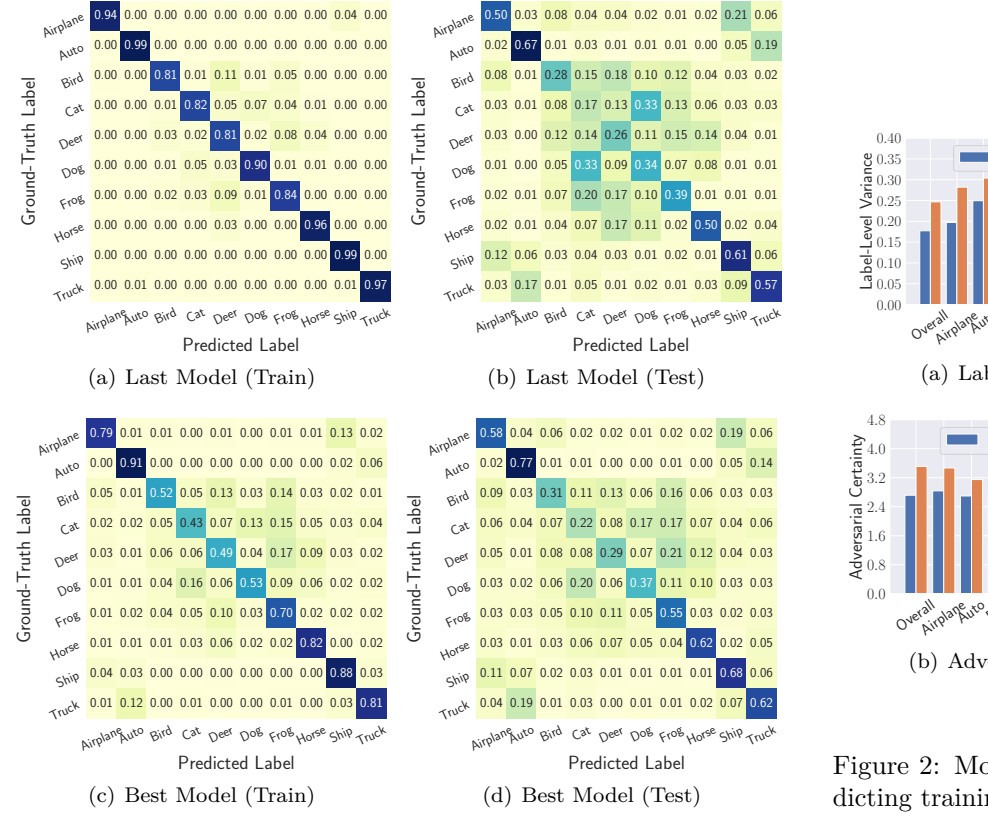

Figure 1: Heatmaps of the label predictions of training- and testing-time generated adversarial examples with respect to models produced from the last and best epochs of adversarial training.

Figure 2: Model confidence in predicting training-time adversarial examples conditioned on the ground-truth class label using different metrics: (a) label-level variance, and (b) adversarial certainty.

et al. (2018); Zhang et al. (2019) suffer from the *robust overfitting* phenomenon Rice et al. (2020): The test-time robustness of intermediate models produced during the training process sharply increases after the first learning rate decay but keeps decreasing afterward. As a result, the model produced from the last training epoch cannot achieve a satisfactory robust generalization performance.

**Heatmap Visualizations.** To gain a deeper understanding of robust overfitting, we visualize the heatmaps of the label predictions for adversarially-perturbed CIFAR-10 images. Given that robust overfitting captures the gap of robust generalization performance with respect to models produced at the last and best epochs, we first plot Figures 1(b) and 1(d). Since only the training process is accessible in adversarial training, we also depict the corresponding training-time heatmaps in Figures 1(a) and 1(c). Here, the ground-truth label represents the underlying class of clean images and the predicted label denotes the class of adversarial examples predicted by the corresponding model. More experimental details about Figure 1 are provided in Appendix B. Specifically, when comparing Figures 1(a) and 1(c), we find that the predictions of *Last Model* mainly concentrate on the ground-truth class, which means the model is overconfident in predicting adversarial examples generated by itself. In contrast, the heatmap of *Best Model*, which achieves better robust generalization performance, depicts less overconfidence. Moreover, by comparing the same model between the training and testing time, i.e., Figure 1(a) versus Figure 1(b) and Figure 1(c) versus Figure 1(d), we discover that the train-test gap is significantly smaller with respect to the Best Model. We note that this result is aligned with the classical machine learning theory: If the testing distribution deviates more from the training distribution, standard learners will show a decreased generalization performance.

According to the above findings, we hypothesize that the overconfidence property is detrimental to robust generalization. To be more specific, if the model cannot generate perturbed training inputs with sufficient uncertainty, the model will not be able to well predict the less certain adversarial examples during the

inference time. Thus, mitigating model overconfidence could be a potential solution to improve robust generalization for adversarial training. In Section 4, we will introduce a novel notion of *Adversarial Certainty*, which is proposed to measure the degree of model overconfidence and is essential for designing our DAC method to help robust generalization as demonstrated in Section 5.

## 4   Introducing Adversarial Certainty

To numerically summarize our findings from the heatmaps, we measure the variance of the class probabilities of the predicted labels, denoted as *label-level variance*, with respect to the training-time adversarial examples for each ground-truth category in Figure 2(a). A lower label-level variance indicates the prediction confidences of different labels are closer, which corresponds to less certainty. More specifically, we observe that the *Best Model* with better robust generalization performance exhibits a lower label-level variance than that of the *Last Model*, which is consistent with the results illustrated in Figure 1. Even though the label-level variance can characterize how certain the training-time generated adversarial examples are, such statistics are on a class level, which is not easy for optimization. Thus, we propose the following logit-level definition, termed as *adversarial certainty*, to capture the certainty of a model in classifying the adversarial examples generated by itself, where a lower score of adversarial certainty suggests the model has a stronger ability to generate less certain adversarial inputs:

**Definition 1** (Adversarial Certainty). Let $\mathcal{X}$ be the input space and $\mathcal{Y} = \{1, 2, \ldots, m\}$ be the label space. Suppose $\mu$ is the underlying distribution and $\mathcal{S}$ is a set of sampled examples. Let $\epsilon \geq 0$, $\Delta$ be the perturbation metric. For any $f_\theta : \mathcal{X} \rightarrow \mathcal{Y}$, we define the *adversarial certainty* of $f_\theta$ as:

$$\mathrm{AC}_\epsilon(f_\theta; \hat{\mu}_\mathcal{S}, \mathcal{A}) = \frac{1}{|\mathcal{S}|} \sum_{(\boldsymbol{x}, y) \in \mathcal{S}} \mathrm{Var}\big(F_\theta\big[\mathcal{A}(\boldsymbol{x}; y, f_\theta, \epsilon)\big]\big),$$

where $\mathcal{A}$ denotes an attack method such as PGD attacks for generating adversarial examples, $F_\theta : \mathcal{X} \rightarrow \mathbb{R}^m$ represents the mapping from the input space $\mathcal{X}$ to the logit layer of $f_\theta$, and $\mathrm{Var}(\boldsymbol{u}) = \sum_{k \in [m]} (u_k - \overline{u})^2 / m$, with $u_k$ and $\overline{u}$ denoting the $k$-th element and mean of $\boldsymbol{u} \in \mathbb{R}^m$ respectively.

Different from label-level variance, adversarial certainty is an averaged sample-wise metric, which calculates the variance of the logits returned by the model $f_\theta$ for each adversarially-perturbed example $\mathcal{A}(\boldsymbol{x}; y, f_\theta, \epsilon)$. Similar to Figure 2(a), we visualize the adversarial certainty of the *Best Model* and the *Last Model* in Figure 2(b). Since predicted labels are decided by the class with the highest predicted probabilities, adversarial certainty depicts a similar pattern to the label-level variance as expected. Based on Definition 1, our hypothesis can then be specifically formulated as:

*Decreasing adversarial certainty during adversarial training can improve robust generalization.*

We note that there also exist other alternative metrics, such as confidence and entropy, which can capture a model's certainty in predicting adversarial examples and summarize the observations of the heatmaps depicted in Figure 1. As will be discussed in Section 6.1, we choose logit-level variance as the metric to define adversarial certainty, mainly because our DAC method illustrated in Section 5 always achieves the best robust generalization performance with such a choice.

**Theoretical Analysis.** To better understand the proposed definition of adversarial certainty, we further study its connection with robust generalization using synthetic data distributions. Following existing works Tsipras et al. (2019); Wei et al. (2023), we assume the following data generating procedure for any example $(\boldsymbol{x}, y) \sim \mu$: The binary label $y$ is first sampled uniformly from $\mathcal{Y} = \{-1, +1\}$, then the robust feature $x_1 = y$ with sampling probability $p$ and $x_1 = -y$ otherwise, while the remaining non-robust features $x_2, \cdots, x_{d+1}$ are sampled i.i.d. from the Gaussian distribution $\mathcal{N}(\eta y, 1)$. Here, $p \in (1/2, 1)$ and $\eta < 1/2$ is a small positive number. Following Wei et al. (2023), we consider linear SVM classifiers: $f_w(\boldsymbol{x}) = \mathrm{sgn}(x_1 + \frac{x_2 + \cdots + x_{d+1}}{w})$ with $w > 0$, where $\mathrm{sgn}(\cdot)$ denotes the sign operator. Subsequently, we assume all the adversarial examples $\boldsymbol{x}'$ are sampled from the following adversarial distribution $\mu_{\mathrm{adv}}(\varepsilon)$ with $\varepsilon > 0$: $x_1' = x_1$, and $x_2', \cdots, x_{d+1}' \overset{i.i.d.}{\sim} \mathcal{N}\big((\eta - \varepsilon)y, 1\big)$. Detailed discussions about the configurations of this synthetic robust classification task are provided in Appendix C.

The following theorem, proven in Appendix C.1, characterizes a connection between the certainty of adversarial examples and the robust generalization performance of an SVM classifier after a single step of gradient update.

**Theorem 1.** Consider the aforementioned data distribution $\mu$ and robust classification task. Let $\varepsilon_{te} \in (\eta, 2\eta)$ and $f_w$ be an arbitrary SVM classifier with $w > 0$. For any $\varepsilon \in [\eta - \frac{w}{d}, \eta]$, $\mathrm{AC}_\varepsilon(f_w; \mu, \mu_{\mathrm{adv}}(\varepsilon))$, the adversarial certainty of $f_w$, is monotonically decreasing with respect to $\varepsilon$. Suppose we conduct one-step gradient update on $w$ using adversarial examples sampled from $\mu_{\mathrm{adv}}(\varepsilon)$: $\hat{w} = w + \alpha \cdot \nabla_w \mathcal{R}(f_w; \mu_{\mathrm{adv}}(\varepsilon))$, where $\alpha > 0$ stands for the learning rate. Then, $\mathcal{R}(f_{\hat{w}}; \mu_{\mathrm{adv}}(\varepsilon_{te}))$, the robust generalization performance of $f_{\hat{w}}$, also increases as $\varepsilon$ increases.

Note that, since we consider the adversarial data distribution $\mu_{\mathrm{adv}}$ instead of $\ell_p$ perturbations, we now generalize the notion of adversarial certainty and robust generalization correspondingly. Theorem 1 suggests that if we decrease the certainty of the adversarial examples sampled from $\mu_{\mathrm{adv}}(\varepsilon)$, the robustness of the SVM classifier $f_{\hat{w}}$ will increase after one-step gradient update based on the sampled adversarial examples, confirming the importance of less certain adversarial examples for robust generalization. We remark that our theoretical analysis can also be extended to the typical setting of $\ell_\infty$-norm bounded perturbations. In Appendix C.2, we show that considering $\ell_\infty$ perturbations is equivalent to considering the adversarial data distribution of $x'_1 = x_1 - y\varepsilon$ and $x'_2, \cdots, x'_{d+1}$ i.i.d. sampled from $\mathcal{N}((\eta - \varepsilon)y, 1)$ for any $w > 0$, and derive similar results to Theorem 1.

## 5 Decreasing Adversarial Certainty Helps Robust Generalization

Previous sections illustrate why decreasing the certainty of adversarial inputs used for adversarial training is beneficial for robust generalization. To further validate our hypothesis, this section proposes a novel method to explicitly **D**ecrease **A**dversarial **C**ertainty (DAC) based on adversarial training. In particular, DAC is designed to find less certain adversarial examples that are used to improve robust generalization, which aims to solve the following optimization problem:

$$\min_{\theta \in \Theta} \frac{1}{|\mathcal{S}_{tr}|} \sum_{(\boldsymbol{x},y) \in \mathcal{S}_{tr}} \max_{\boldsymbol{x}' \in \mathcal{B}_\epsilon(\boldsymbol{x})} L\big(f_{\theta'}, \boldsymbol{x}', y\big), \text{ where } \theta' = \operatorname*{argmin}_{\theta' \in \mathcal{C}(\theta)} \mathrm{AC}_\epsilon(f_\theta; \mathcal{S}_{tr}, \mathcal{A}). \tag{2}$$

$\mathcal{S}_{tr}$ is the clean training dataset, $\mathcal{A}$ denotes a specific attack method (e.g., PGD attacks $\mathcal{A}_{\mathrm{pgd}}$), and $\mathcal{C}(\theta)$ represents the feasible search region for $\theta'$. We remark that imposing the constraint of $\mathcal{C}(\theta)$ is necessary, because the goal of DAC is to improve robust generalization of adversarial training, instead of merely obtaining adversarial certainty as low as possible. Without such a constraint, minimizing adversarial certainty will cause $\theta'$ to significantly deviate from the initial $\theta$. This will render the adversarial examples generated with respect to $\theta'$ less useful, thereby inducing a negative impact on robust generalization (see Figure 3(a) and our correlation analysis for more discussions regarding the design choice of imposing such a constraint set).

Directly solving the min-max-min problem introduced in Equation (2) is challenging, due to the non-convex nature of the optimization and the implicit definition of $\mathcal{C}(\theta)$. Thus, we resort to gradient-based methods for an approximate solver. To be more specific, we take the $t$-th iteration of adversarial training as an example to illustrate our design of DAC. Given a set of clean training examples $\mathcal{S}_{tr}$, a specific attack method $\mathcal{A}$, and a classification model $f_\theta$, our DAC method can be formulated as a two-step optimization:

$$\begin{aligned}
\theta_{t+0.5} &= \theta_t - \lambda \cdot \nabla_\theta \mathrm{AC}_\epsilon(f_\theta; \mathcal{S}_{tr}, \mathcal{A})\Big|_{\theta=\theta_t}, \\
\theta_{t+1} &= \theta_{t+0.5} - \gamma \cdot \nabla_\theta L_{\mathrm{rob}}(f_\theta; \mathcal{S}_{tr}, \mathcal{A})\Big|_{\theta=\theta_{t+0.5}},
\end{aligned} \tag{3}$$

where $\lambda > 0$ and $\gamma > 0$ represent the step sizes of the two optimization steps, $\mathrm{AC}_\epsilon(f_\theta; \mathcal{S}_{tr}, \mathcal{A})$ denotes the adversarial certainty of $f_\theta$ with respect to $\mathcal{S}_{tr}$ and $\mathcal{A}$, and $L_{\mathrm{rob}}(f_\theta; \mathcal{S}_{tr}, \mathcal{A})$ can be roughly understood as the robust loss except that the inner maximization is approximated using some attack method such as $\mathcal{A}_{\mathrm{pgd}}$. The first step in Equation (3) optimizes the adversarial certainty, which adjusts the model parameters $\theta_t$ in a direction such that the generated training-time adversarial examples are less certain, whereas the second step in Equation (3) optimizes the model's ability in distinguishing adversarial examples generated by the model itself as in standard adversarial training.

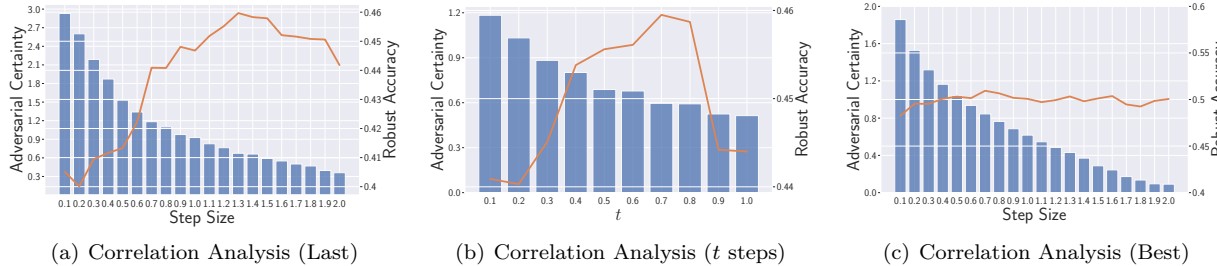

(a) Correlation Analysis (Last)  (b) Correlation Analysis ($t$ steps)  (c) Correlation Analysis (Best)

Figure 3: Correlation between adversarial certainty and robust generalization under different configurations.

**Correlation Analysis.** Since our work aims to improve robust generalization by finding less certain adversarial examples, it is natural to ask the following question:

*Does decreasing adversarial certainty always induce better robust generalization?*

Recall that in Equation (2), $\mathcal{C}(\theta)$ defines the feasible region for optimizing adversarial certainty. Therefore, the answer would be affirmative within this region, i.e., decreasing adversarial certainty will increase test robust accuracy. To support the answer to this question with evidence, we conduct a correlation analysis between adversarial certainty and robust generalization. The results are illustrated in Figure 3(a). Specifically, we use an AT-trained model as the starting point, from which the heatmaps in Figure 1 are derived. Then, we respectively update the model with one more epoch using DAC with different step sizes, ranging from 0.1 to 2.0, in the $\theta_t \rightarrow \theta_{t+0.5}$ step of Equation (2) to decrease adversarial certainty. Finally, we measure the training-time adversarial certainty (i.e., the blue bars) and robust test accuracy (i.e., the orange curve) of the result models. Figure 3(a) shows that adversarial certainty keeps decreasing as the step size increases. Meanwhile, the model's robustness first keeps improving but then decreases when the step size is beyond the value of 1.3. This result suggests that if the model parameters lie in the feasible search region with a properly selected step size, lower adversarial certainty leads to higher test robust accuracy. However, when the model is out of the feasible search region, decreasing adversarial certainty will no longer improve robust generalization.

The above investigation suggests a negative answer to the question by performing one-epoch optimization with different step sizes on the last model of AT, i.e., the model of Figures 1(a) and 1(b). Here, we demonstrate that our finding is indeed derived from the changes of adversarial certainty but not from the one-epoch setting. To be more specific, we separate one step of decreasing adversarial certainty with size 0.7 into $t$ steps, i.e., each step corresponds to the size of $0.7/t$, where $t = \{1, 2, 3, 4, 5, 6, 7, 8, 9, 10\}$. The results of training-time adversarial and testing-time robust generalization are depicted in Figure 3(b), where we can observe a consistent pattern with Figure 3(a) that robust generalization first gains more improvements but then less with the decrease of adversarial certainty.

Moreover, we apply the one-epoch optimization on the best AT model that corresponds to Figures 1(c) and 1(d) to derive more observations. From Figure 3(c), we can find that the changes of robust generalization with the increase of adversarial certainty depict a different pattern from Figure 3(a). First, the robust generalization improvements are slighter than those in the last model. Besides, there is no trend of robust generalization decreasing even if the step size arrives at 2.0. These results suggest that the feasible region of the best model is larger than that of the last model, which suffers from more severe robust overfitting. Consequently, when applying our DAC method, an overconfident model is supposed to select the optimization step size carefully; otherwise, the selection could be more ambitious.

## 6  Experiments

This section examines the performance of our DAC method under $\ell_\infty$ perturbations with $\epsilon = 8/255$ on various model architectures, including PreActResNet-18, denoted as PRN18, and WideResNet-34, denoted as WRN34. And we train a model for 200 epochs using SGD with a momentum of 0.9. Besides, the initial learning rate is 0.1, and is divided by 10 at the 100-th epoch and at the 150-th epoch. The adversarial attack used in training is PGD-10 with a step size of 1/255 for SVHN, and 2/255 for CIFAR-10 and CIFAR-100, while we

Table 1: Testing-time robustness (%) with/without DAC on CIFAR-10 under $\ell_\infty$ perturbations across different architectures and adversarial training methods. The best performance is highlighted in bold.

| Architecture | Method | Clean | PGD-20 | PGD-100 | $CW_\infty$ | AutoAttack |
|---|---|---|---|---|---|---|
| PRN18 | AT | 82.88 (82.68) | 41.51 (49.23) | 40.96 (48.92) | 41.61 (48.07) | 39.66 (45.71) |
| | + DAC | **84.64 (83.55)** | **45.55 (52.20)** | **44.94 (51.87)** | **44.55 (50.05)** | **42.78 (48.20)** |
| | TRADES | 82.10 (81.33) | 47.44 (51.65) | 46.95 (51.42) | 46.64 (49.18) | 44.99 (48.06) |
| | + DAC | **83.18 (82.80)** | **49.32 (52.90)** | **48.81 (52.67)** | **48.30 (50.11)** | **46.40 (48.96)** |
| | MART | 80.85 (78.27) | 50.23 (52.28) | 49.71 (52.13) | 46.88 (47.83) | 44.68 (46.01) |
| | + DAC | **81.12 (79.37)** | **52.38 (53.25)** | **52.04 (53.14)** | **48.97 (49.25)** | **47.24 (47.69)** |
| WRN34 | AT | 86.47 (**85.86**) | 47.25 (55.31) | 46.73 (55.00) | 47.85 (54.04) | 45.84 (51.94) |
| | + DAC | **86.48** (85.10) | **52.02 (57.93)** | **51.69 (57.68)** | **51.51 (54.98)** | **49.75 (53.33)** |
| | TRADES | 83.37 (81.40) | 51.51 (58.78) | 51.28 (58.72) | 49.26 (53.33) | 47.74 (52.63) |
| | + DAC | **85.04 (84.55)** | **58.97 (60.96)** | **58.97 (60.81)** | **52.79 (55.00)** | **51.80 (53.99)** |
| | MART | 83.11 (**83.30**) | 48.93 (58.13) | 48.31 (57.75) | 46.32 (52.22) | 44.89 (50.31) |
| | + DAC | **84.69** (80.09) | **52.00 (59.31)** | **51.32 (59.26)** | **49.50 (53.02)** | **47.65 (51.48)** |

utilize the commonly-used attack benchmarks of PGD-20 Madry et al. (2018), PGD-100 Madry et al. (2018), $CW_\infty$ Carlini & Wagner (2017) and AutoAttack Croce & Hein (2020) for evaluation. In addition, we measure the *Clean* performance to investigate the influence on clean images. Regarding other hyperparameters, we follow the settings described in their original papers. In all cases, we evaluate the performance of the last (best) model in terms of testing-time robust accuracy.

In Section 6.1, we evaluate the effectiveness of our DAC method in improving robust generalization on three widely-used benchmark datasets: CIFAR-10 Krizhevsky & Hinton (2009), CIFAR-100 Krizhevsky & Hinton (2009) and SVHN Netzer et al. (2011) based on three baseline adversarial training methods: AT Madry et al. (2018), TRADES Zhang et al. (2019) and MART Wang et al. (2020). To study the generalizability of our method, we further conduct experiments under $\ell_2$ perturbations, where we set $\epsilon = 128/255$ with a step size of $15/255$ for all datasets. In Section 6.2, we associate with other robustness-enhancing techniques to further investigate the effect of adversarial certainty in adversarial training. Finally, we demonstrate the efficacy of DAC under a simplified one-step optimization setting in Section 6.3, and improve the DAC efficiency by regularizing decreasing adversarial certainty in a loss term, i.e., DAC_Reg that will be detailed in Section 6.4.

## 6.1 Main Results

We first evaluate the robust generalization performance of our proposed DAC method on the benchmark CIFAR-10 image dataset. The comparison results are depicted in Table 1, showing that DAC significantly enhances model robustness across different adversarial attacks, such as PGD attacks Madry et al. (2018), CW attacks Carlini & Wagner (2017) and AutoAttack Croce & Hein (2020). These results demonstrate the effectiveness of DAC, indicating the significance of generating less certain adversarial examples for robust generalization. Besides, we observe that although WRN34 suffers from more severe robust overfitting using baseline adversarial training methods, it achieves more robustness improvement by our method. This suggests that WRN34 is superior to PRN18 in terms of robust generalization with the help of DAC. In addition to adversarial robustness, it is also worth noting the effect of DAC on clean test accuracy, which captures the standard generalization ability of the model. Table 1 reveals that DAC consistently improves the clean test accuracy under all experimental settings. This promotion shows that DAC could also help models gain better generalization performance on unseen clean images even by learning from adversarial examples. The results that include evaluations on more benchmark datasets (i.e., SVHN and CIFAR-100) are depicted in Tables 2 and 3, of which the full versions are shown in Tables 7 and 8 (Appendix E) due to space limit, showing a similar pattern of improvements.

Moreover, we empirically study the impact of DAC on the phenomenon of robust overfitting. More specifically, we evaluate the gap of testing-time adversarial robustness between the best and the last models. The results

Table 2: Testing-time adversarial robustness (%) of AT on SVHN with/without DAC/DAC_Reg under $\ell_\infty$ perturabtions across different model architectures and benchmark datasets. The complete results are shown in Table 7, including all dataset settings of SVHN, CIFAR-10 and CIFAR-100.

| Dataset | Architecture | Method | Clean | PGD-20 | PGD-100 | CW$_\infty$ | AutoAttack |
|---|---|---|---|---|---|---|---|
| SVHN | PRN18 | AT | 89.63 (88.64) | 42.25 (51.00) | 41.37 (50.30) | 42.84 (48.19) | 39.52 (46.02) |
| | | + DAC | 90.58 (89.63) | **45.86** (**54.42**) | **43.92** (**53.78**) | **43.75** (**50.15**) | 40.68 (**48.23**) |
| | | + DAC_Reg | **90.65** (**90.21**) | 45.39 (53.06) | 43.77 (52.28) | 43.66 (49.64) | **41.10** (47.39) |
| | WRN34 | AT | 91.51 (89.72) | 46.81 (53.43) | 44.94 (52.77) | 45.76 (50.43) | 41.71 (49.50) |
| | | + DAC | 91.26 (91.83) | 60.42 (**67.95**) | 56.71 (**64.85**) | 56.98 (**65.09**) | 42.33 (**50.42**) |
| | | + DAC_Reg | **91.76** (**92.13**) | **62.19** (65.96) | **59.54** (63.68) | **60.05** (63.87) | **42.46** (49.95) |

Table 3: Testing-time adversarial robustness (%) of AT, TRADES and MART with/without DAC on SVHN under $\ell_\infty$ perturbations. The complete results are shown in Table 8, including all dataset settings of SVHN, CIFAR-10 and CIFAR100.

| Dataset | Method | Clean | PGD-20 | PGD-100 | CW$_\infty$ | AutoAttack |
|---|---|---|---|---|---|---|
| SVHN | AT | 89.63 (88.64) | 42.25 (51.00) | 41.37 (50.30) | 42.84 (48.19) | 39.52 (46.02) |
| | + DAC | 90.58 (89.63) | **45.86** (**54.42**) | **43.92** (**53.78**) | **43.75** (**50.15**) | 40.68 (**48.23**) |
| | + DAC_Reg | **90.65** (**90.21**) | 45.39 (53.06) | 43.77 (52.28) | 43.66 (49.64) | **41.10** (47.39) |
| | TRADES | 89.12 (87.75) | 51.50 (55.19) | 50.69 (54.50) | 45.50 (50.32) | 45.02 (48.69) |
| | + DAC | **90.24** (89.59) | **52.24** (**57.09**) | **51.14** (**56.39**) | **46.34** (**52.22**) | **46.20** (**50.52**) |
| | + DAC_Reg | 90.03 (**89.75**) | 51.78 (56.10) | 50.92 (54.83) | 45.86 (51.35) | 45.30 (49.06) |
| | MART | 89.68 (84.48) | 49.07 (52.30) | 48.30 (52.22) | 45.48 (48.04) | 44.54 (47.38) |
| | + DAC | 88.90 (84.64) | **51.04** (**53.64**) | **50.91** (**52.70**) | **46.94** (**49.96**) | **46.18** (**48.50**) |
| | + DAC_Reg | **90.18** (**88.47**) | 50.94 (52.94) | 49.87 (52.46) | 46.32 (49.18) | 45.86 (47.73) |

Table 4: Testing-time adversarial robustness (%) of AT with/without DAC on PreActResNet-18 under $\ell_2$ perturabtions against PGD-20 across different benchmark datasets.

| Method | SVHN | | CIFAR-10 | | CIFAR-100 | |
|---|---|---|---|---|---|---|
| | Best | Last | Best | Last | Best | Last |
| AT | 66.45 | 63.20 | 66.02 | 65.18 | 39.23 | 35.68 |
| + DAC | **69.11** | **67.44** | **69.10** | **67.37** | **40.75** | **36.32** |

are shown in Figure 4(a), where DAC consistently mitigates robust overfitting across different settings. These results indicate that decreasing adversarial certainty can successfully mitigate robust overfitting. Besides, we also measure the adversarial certainty gap between the best model and the last model produced by AT and AT-DAC in Figure 4(b). It can be observed that the adversarial certainty gap of AT-DAC is significantly smaller than that of AT, which is aligned with the closer adversarial robustness of the best model and the last model.

**Comparison with Other Metrics.** Recall our discussions in Section 3, we propose the notion of adversarial certainty based on logit-level variance (Definition 1), which is further used in our design of DAC. Noticing that confidence and entropy are also relevant metrics that can capture the model's overconfidence in predicting adversarial examples, we conduct a case study to illustrate why we choose to define adversarial certainty based on variance. For ease of presentation, we only present results on CIFAR-10 and AT as an illustration, where similar trends are observed among other settings. Table 5 reports the test-time adversarial robustness of models learned using AT-DAC with different metrics used in the definition of adversarial certainty. We can see that the last and best models produced using our method with the variance metric achieve the best robustness performance, which empirically supports our design choice.

$\ell_2$-**Norm Bounded Perturbations.** In the above evaluation, we focus on the $\ell_\infty$ norm-bounded perturbations. Meanwhile, the $\ell_2$ norm is also a prevalent perturbation setting in adversarial training. Thus, in

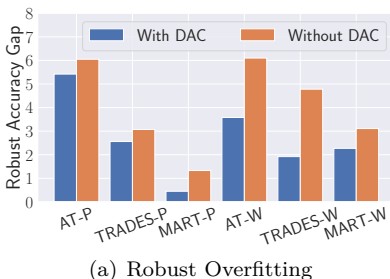 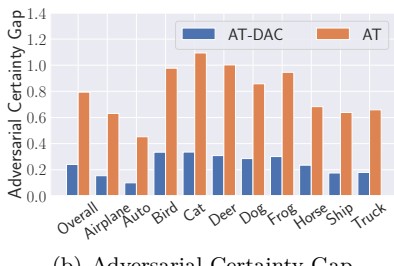 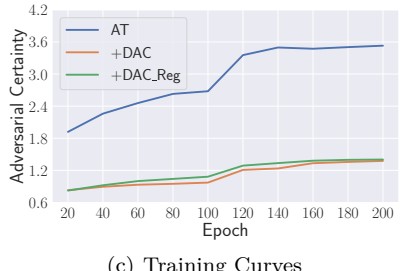

|  (a) Robust Overfitting | (b) Adversarial Certainty Gap | (c) Training Curves |

Figure 4: (a) Robust overfitting across different methods, where "-P" and "-W" represent PRN18 and WRN34 respectively. (b) Adversarial certainty gap with respect to AT and AT-DAC conditioned on different ground-truth classes. (c) Training curves of adversarial certainty with respect to different adversarial training algorithms.

Table 5: Comparison results (%) of different metrics defining adversarial certainty on PRN18 and CIFAR-10 at the last and best epochs.

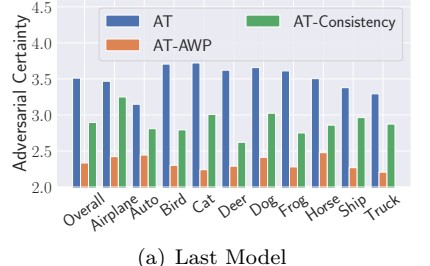 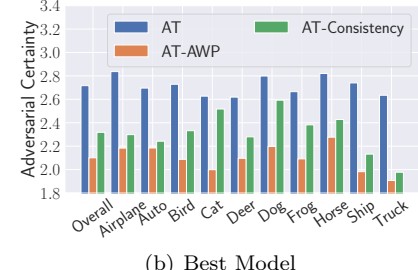

| (a) Last Model | (b) Best Model |

|            | Last      | Best      |
| ---------- | --------- | --------- |
| Confidence | 44.40     | 51.14     |
| Entropy    | 44.27     | 51.00     |
| Variance   | **45.55** | **52.20** |

Figure 5: Adversarial certainty across different CIFAR-10 classes with on the last and best models.

Table 4, we evaluate our method under the $\ell_2$ perturbations. Similarly, DAC depicts consistent improvements in adversarial robustness on best and last epochs across different benchmark datasets, which shows the efficacy of DAC against adversarial attacks with $\ell_2$ perturbations.

## 6.2 Effect of Adversarial Certainty on Other Robustness-Enhancing Techniques

We note that several recent works also focus on understanding robust generalization and developing methods to improve adversarial training, including adversarial weight perturbation Wu et al. (2020) (AWP), and consistency regularization Tack et al. (2022) (Consistency). More concretely, Wu et al. discovered that the flatness of the weight loss landscape is an important factor related to robust generalization Wu et al. (2020). And the method of Consistency regularizes the adversarial consistency based on various data augmentations Tack et al. (2022). However, since these methods focus on different strategies to improve robust generalization, it is unclear whether our proposed adversarial certainty has any connection with them. Therefore, we study the changes in adversarial certainty when involving AWP and Consistency in adversarial training, respectively, which are shown in Figure 5. Surprisingly, we find that AWP and Consistency, which improve the robust generalization of AT on both the last and best models, can gain lower adversarial certainty. These findings are consistent with the idea behind our DAC method – decreasing adversarial certainty helps robust generalization. In other words, AWP and Consistency, which are designed toward their specified directions, will implicitly decrease adversarial certainty. Note that, even if AWP and Consistency have influences on adversarial certainty, it does not mean that our work proposes a similar concept to them. Specifically, adversarial certainty is derived by observing an adversarial-training-unique phenomenon – robust overfitting, meanwhile, AWP is inspired by the theory of weight loss landscape from standard learning and Consistency considers the augmentation scope. Consequently, our proposed adversarial certainty is a crucial property in adversarial training, which can either explicitly or implicitly affect robust generalization.

As AWP and Consistency can implicitly improve adversarial certainty, we then investigate the compatibility of our DAC method with AWP and Consistency by a naive attempt. To incorporate DAC in AWP, we add a step

Table 6: Testing-time adversarial robustness (%) of AWP and Consistency with/without DAC on CIFAR-10 and PRN18 under $\ell_\infty$ perturabtions.

| Method | Clean | PGD-100 | CW$_\infty$ | AutoAttack |
|---|---|---|---|---|
| AT-AWP | 83.76±0.06 (82.37±0.07) | 52.71±0.26 (53.89±0.27) | 51.07±0.24 (51.22±0.24) | 48.75±0.23 (49.33±0.27) |
| + DAC | **84.07±0.13 (82.67±0.10)** | **54.30±0.26 (55.00±0.31)** | **51.76±0.25 (52.03±0.22)** | **49.80±0.26 (49.96±0.20)** |
| TRADES-AWP | 81.46±0.13 (81.28±0.08) | 52.54±0.31 (53.55±0.26) | 50.37±0.23 (50.61±0.21) | 49.54±0.25 (49.92±0.23) |
| + DAC | **82.69±0.06 (82.85±0.08)** | **53.80±0.29 (54.49±0.29)** | **51.44±0.23 (51.53±0.21)** | **50.51±0.26 (50.63±0.25)** |
| MART-AWP | 78.13±0.06 (77.27±0.09) | 53.06±0.25 (52.58±0.31) | 49.05±0.28 (48.39±0.22) | 46.53±0.22 (47.01±0.26) |
| + DAC | **80.03±0.06 (78.65±0.11)** | **54.67±0.29 (54.93±0.30)** | **49.58±0.25 (49.14±0.21)** | **47.47±0.27 (47.73±0.21)** |
| AT-Consistency | 85.28±0.06 (84.66±0.08) | 55.16±0.31 (56.46±0.27) | 50.81±0.23 (51.13±0.21) | 48.08±0.21 (48.48±0.23) |
| + DAC | **85.36±0.09 (85.17±0.13)** | **56.31±0.26 (56.90±0.26)** | **51.29±0.27 (51.72±0.22)** | **49.00±0.21 (49.46±0.25)** |
| TRADES-Consistency | 83.68±0.12 (83.51±0.08) | 52.78±0.26 (52.79±0.31) | 48.85±0.21 (48.89±0.28) | 47.75±0.21 (47.77±0.20) |
| + DAC | **84.78±0.12 (84.73±0.06)** | **53.48±0.27 (53.72±0.26)** | **49.37±0.27 (49.41±0.28)** | **48.15±0.21 (48.19±0.21)** |
| MART-Consistency | 78.21±0.10 (78.11±0.09) | 56.31±0.29 (56.81±0.28) | 47.33±0.28 (47.47±0.21) | 45.53±0.27 (45.73±0.22) |
| + DAC | **81.91±0.06 (81.35±0.10)** | **58.29±0.33 (58.56±0.28)** | **50.08±0.27 (50.21±0.27)** | **48.28±0.22 (48.59±0.27)** |

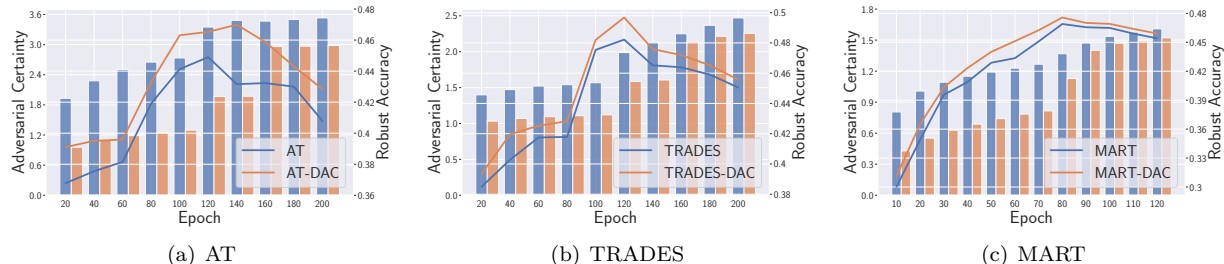

| (a) AT | (b) TRADES | (c) MART |
|---|---|---|

Figure 6: Visualization results for comparing the adversarial certainty and robust generalization of different adversarial training methods with and without the involvement of DAC.

before weight perturbation to optimize the certainty of adversarial examples. Then the updated intermediate model is used to generate new adversarial examples for the following AWP optimization. Similarly, we first explicitly update the adversarial certainty on augmented samples, and then follow the Consistency optimization. The results are shown in Table 6. As expected, since AWP and Consistency have already implicitly decreased adversarial certainty, even if DAC conducts an explicit optimization, our method can only gain limited benefit. Nevertheless, our repeated trials demonstrate that the improvements, even slight, are indeed derived from our method rather than randomness. Further, we conduct a significance test, which shows that the improvements of robust generalization on AWP and Consistency are statistically significant, as fully presented in Appendix D. The goal of our work is to propose adversarial certainty and clarify its significance in adversarial training, thus better designs of involving adversarial certainty in existing robustness-enhancing strategies are left as future work.

### 6.3 Further Discussion on DAC

Based on previous results, we demonstrate the benefits of involving our DAC method in adversarial training. To more intuitively demonstrate the efficacy of our method, we empirically measure the performance improvements derived by conducting DAC for a single epoch starting with different models. First, we train a sequence of models by AT and TRADES for 200 epochs, and by MART for 120 epochs, respectively. For every 20 epochs, we then update the same intermediate model by one further epoch using each of the three adversarial training methods with and without the help of DAC. Finally, we measure the adversarial certainty and robust generalization for all the updated models. Figure 6 summarizes the results, where the blue color represents the original method without DAC and orange corresponds to results with our DAC. The bars show adversarial certainty and the curves depict robust generalization. It can be seen from Figure 6 that starting with different intermediate models, DAC can consistently gain less certain adversarial examples, from which the updated model attains better robust generalization performance, which is aligned with our theoretical

results shown in Theorem 1. By optimizing the same model with only one epoch, these comparison results clearly show the efficacy of DAC for adversarially-trained models.

### 6.4 Improvement on DAC Efficiency

To gain a better understanding of our method, we explicitly examine our proposed adversarial certainty by involving two steps for each iteration i.e., DAC, as formulated in Equation (3). In this section, we propose a more efficient method, denoted as DAC_Reg, by regularizing the optimization of adversarial certainty as a term in adversarial training loss. More concretely, the optimization problem with the additional regularizer can be cast as:

$$\min_{\theta \in \Theta} \frac{1}{|\mathcal{S}_{tr}|} \sum_{(\boldsymbol{x}, y) \in \mathcal{S}_{tr}} L\big(f_\theta, \boldsymbol{x}', y\big) + \beta \cdot \mathrm{AC}_\epsilon(f_\theta; \mathcal{S}_{tr}, \mathcal{A}),$$

where $\beta > 0$ denotes the trade-off parameter between the regularization of adversarial certainty and the robust loss. Similar to adversarial training, the model parameters are iteratively updated using stochastic gradient descent (SGD) with respect to the regularized robust loss. Benefiting from the regularization design, DAC_Reg requires similar training time to the standard adversarial learning, which is $0.56\times$ of that of DAC. For instance, for a PRN18 model of AT and CIFAR-10 on a single NVIDIA A100 GPU, DAC averagely costs 143s for each training epoch while DAC_Reg costs 80s. The comparison results of AT, TRADES and MART models on SVHN, CIFAR-10 and CIFAR-100 datasets are shown in Table 2 and Table 3. We can see that DAC_Reg achieves comparable performance, due to the additional penalty on adversarial certainty, which is only a bit inferior to DAC. In a few cases, DAC could bring better and more stable improvements. For instance, when a PRN18 model is trained on CIFAR-100 by AT, DAC_Reg can only gain the improvement on the last epoch but not on the best epoch. In addition, we measure the adversarial certainty of a sequence of models trained by AT, DAC and DAC_Reg, respectively, in Figure 4(c). We observe that DAC gains the lowest adversarial certainty with a slight advantage over DAC_Reg, again indicating that lower adversarial certainty corresponds to higher robust generalization.

## 7 Conclusion and Future Work

We revisited the robust overfitting phenomenon of adversarial training and argued that model overconfidence in predicting training-time adversarial examples is a potential cause. Accordingly, we introduced the notion of adversarial certainty to capture the degree of overconfidence and designed a strategy to decrease adversarial certainty for models produced during adversarial training. Experiments on image benchmarks demonstrate the effectiveness of our method, which confirms the importance of generating less certain adversarial examples for robust generalization. Our work aims to gain a better understanding of robust generalization through observations from robust overfitting. We believe our work provides a significant contribution to advancing the field of adversarial machine learning, which might inspire practitioners to look into the important role of less certain adversarial examples when building real-world robust systems against adversarial examples.

Investigating whether the notion of adversarial certainty connects with the robustness of language models could be an interesting future direction of our work. Nevertheless, natural language processing (NLP) is a different field from computer vision (CV), as it focuses on the discrete space while that of CV is continuous. Thus, directly transferring our method to the NLP domain is non-trivial. According to the pipeline of vanilla adversarial training for text classification (Morris et al., 2020), the generation of adversarial examples is to substitute some selected words. Such a generation scheme is different from that of CV and does not support the gradient-based optimization of decreasing adversarial certainty. In that case, we will need to design an additional objective function to measure how certain an NLP model is about its generated sentences, which could be the metric for selecting and substituting words. We regard the exploration of DAC in other domains like NLP as interesting future work but is beyond the scope of our work. We believe our proposed concept of adversarial certainty will provide important insights for the development of robust machine learning systems.

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

# Appendix

## A   Complete Introduction of Preliminaries

For the sake of completeness, this section presents the detailed definitions and discussions of the preliminary concepts introduced in Section 3, including adversarial robustness, robust generalization and adversarial training. Let $(\mathcal{X}, \Delta)$ be a metric space. For any set $\mathcal{C} \subseteq \mathcal{X}$ and any $\boldsymbol{x} \in \mathcal{X}$, let $\Pi_{\mathcal{C}}(\boldsymbol{x}) = \operatorname{argmin}_{\boldsymbol{x}' \in \mathcal{C}} \Delta(\boldsymbol{x}', \boldsymbol{x})$ be the projection of $\boldsymbol{x}$ onto $\mathcal{C}$.

**Adversarial Robustness.** Adversarial robustness captures the classifier's resilience to small adversarial perturbations. In particular, we work with the following definition of adversarial robustness:

**Definition 2** (Adversarial Robustness). Let $\mathcal{X} \subseteq \mathbb{R}^n$ be input space, $\mathcal{Y}$ be label space, and $\mu$ be the underlying distribution of inputs and labels. Let $\Delta$ be a distance metric on $\mathcal{X}$ and $\epsilon \geq 0$. For any classifier $f_\theta : \mathcal{X} \to \mathcal{Y}$, the *adversarial robustness* of $f_\theta$ with respect to $\mu$, $\epsilon$ and $\Delta$ is defined as:

$$\mathcal{R}_\epsilon(f_\theta; \mu) = 1 - \Pr_{(\boldsymbol{x},y) \sim \mu} \left[ \exists\, \boldsymbol{x}' \in \mathcal{B}_\epsilon(\boldsymbol{x}) \text{ s.t. } f_\theta(\boldsymbol{x}') \neq y \right]. \tag{4}$$

When $\epsilon = 0$, $\mathcal{R}_0(f_\theta; \mu)$ is equivalent to the clean accuracy of $f_\theta$. In practice, the probability density function of the underlying distribution $\mu$ is typically unknown. Instead, we only have access to a set of test examples $\mathcal{S}_{te}$ i.i.d. sampled from $\mu$. Thus, a classifier's adversarial robustness is estimated by replacing $\mu$ in Equation (4) with its empirical counterpart based on $\mathcal{S}_{te}$. To be more specific, the testing-time adversarial robustness of $f_\theta$ with respect to $\mathcal{S}_{te}$, $\epsilon$ and $\Delta$ is given by:

$$\mathcal{R}_\epsilon(f_\theta; \hat{\mu}_{\mathcal{S}_{te}}) = 1 - \frac{1}{|\mathcal{S}_{te}|} \sum_{(\boldsymbol{x},y) \in \mathcal{S}_{te}} \max_{\boldsymbol{x}' \in \mathcal{B}_\epsilon(\boldsymbol{x})} \mathbb{1}\left( f_\theta(\boldsymbol{x}') \neq y \right), \tag{5}$$

where $\hat{\mu}_{\mathcal{S}_{te}}$ denotes the empirical measure of $\mu$ based on $\mathcal{S}_{te}$. We remark that *robust generalization*, the main subject of this study, captures how well a model can classify adversarially-perturbed inputs that are not used for training, which is essentially the testing-time adversarial robustness $\mathcal{R}_\epsilon(f_\theta; \hat{\mu}_{\mathcal{S}_{te}})$. And we write $\mathcal{R}_\epsilon(f_\theta) = \mathcal{R}_\epsilon(f_\theta; \hat{\mu}_{\mathcal{S}_{te}})$ in the following discussions when $\hat{\mu}_{\mathcal{S}_{te}}$ is free of context. In this work, we focus on the $\ell_p$-norm distances as the perturbation metric $\Delta$, since they are most widely-used in existing literature on adversarial examples. Although $\ell_p$ distances may not best reflect the human-perceptual similarity Sharif et al. (2018) and perturbation metrics beyond $\ell_p$-norm such as geometrically transformed adversarial examples Kanbak et al. (2018); Xiao et al. (2018) were also considered in literature, there is still a significant amount of interest in understanding and improving model robustness against $\ell_p$ perturbations. We hope that our insights gained from $\ell_p$ perturbations will shed light on how to learn better robust models for more realistic adversaries.

**Adversarial Training.** Among all the existing defenses against adversarial examples, *adversarial training* Madry et al. (2018); Zhang et al. (2019); Carmon et al. (2019) is most promising in producing robust models. Given a set of training examples $\mathcal{S}_{tr}$ sampled from $\mu$, adversarial training aims to solve the following min-max optimization problem:

$$\min_{\theta \in \Theta} L_{\mathcal{R}}(f_\theta; \mathcal{S}_{tr}), \text{ where } L_{\mathcal{R}}(f_\theta; \mathcal{S}_{tr}) = \frac{1}{|\mathcal{S}_{tr}|} \sum_{(\boldsymbol{x},y) \in \mathcal{S}_{tr}} \max_{\boldsymbol{x}' \in \mathcal{B}_\epsilon(\boldsymbol{x})} L\left( f_\theta, \boldsymbol{x}', y \right). \tag{6}$$

Here, $\Theta$ denotes the set of model parameters, and $L$ is typically set as a convex surrogate loss such that $L\left( f_\theta, \boldsymbol{x}, y \right)$ is an upper bound on the 0-1 loss $\mathbb{1}\left( f_\theta(\boldsymbol{x}) \neq y \right)$ for any $(\boldsymbol{x}, y)$. For instance, $L$ is set as the cross-entropy loss in vanilla adversarial training Madry et al. (2018), whereas the combination of a cross-entropy loss for clean data and a regularization term for robustness is used in TRADES Zhang et al. (2019). In theory, if $\mathcal{S}_{tr}$ well captures the underlying distribution $\mu$ and the robust loss $L_{\mathcal{R}}(f_\theta; \mathcal{S}_{tr})$ is sufficiently small, then $f_\theta$ is guaranteed to achieve high adversarial robustness $\mathcal{R}_\epsilon(f_\theta; \mu)$.

However, directly solving the min-max optimization problem (6) for non-convex models such as deep neural networks is challenging. It is typical to resort to some good heuristic algorithm to approximately solve the

problem, especially for the inner maximization problem. In particular, Madry et al. proposed to alternatively solve the inner maximization using an iterative projected gradient descent method (PGD) and solve the outer minimization using SGDMadry et al. (2018), which is regarded as the go-to approach in the research community. We further explain its underlying mechanism below. For any intermediate model $f_\theta$ produced during adversarial training, PGD updates the (perturbed) inputs according to the following update rule:

$$\boldsymbol{x}_{s+1} = \Pi_{\mathcal{B}_\epsilon(\boldsymbol{x})}\big(\boldsymbol{x}_s + \alpha \cdot \mathrm{sgn}(\nabla_{\boldsymbol{x}_s} L(f_\theta, \boldsymbol{x}_s, y))\big) \text{ for any } (\boldsymbol{x}, y) \text{ and } s \in \{0, 1, \dots, S-1\}, \tag{7}$$

where $\boldsymbol{x}_0 = \boldsymbol{x}$, $\alpha > 0$ denotes the step size and $S$ denotes the total number of iterations. For the ease of presentation, we use $\mathcal{A}_{\mathrm{pgd}}$ to denote PGD attacks such that for any example $(\boldsymbol{x}, y)$ and classifier $f_\theta$, it generates $\boldsymbol{x}' = \boldsymbol{x}_S = \mathcal{A}_{\mathrm{pgd}}(\boldsymbol{x}; y, f_\theta, \epsilon)$ based on the update rule (7). After generating the perturbed input for each example in a training batch, the model parameter $\theta$ is then updated by a single SGD step with respect to $L(f_\theta, \boldsymbol{x}', y)$ for the outer minimization problem in Equation (6).

## B  More Details of Figures in Sections 3 and 4

This section provides all the experimental details for producing the heatmaps and the histograms illustrated in Sections 3 and 4. Given a model $f_\theta$ (e.g., *Best Model* and *Last Model*) and a set of examples $\mathcal{S}$ sampled from the underlying distribution $\mu$ (e.g., CIFAR-10 training and testing datasets), adversarial examples are generated by PGD attacks within the perturbation ball $\mathcal{B}_\epsilon(\boldsymbol{x})$ centered at $\boldsymbol{x}$ with radius $\epsilon = 8/255$ under the $\ell_\infty$ perturbations, which follows the settings of generating training samples considered in Section 6, e.g., PGD is iteratively conducted by 10 steps with the step size of $2/255$. We record and plot the label predictions of the generated adversarial examples with respect to each model as heatmaps in Figure 1.

Let HM be the $m \times m$ matrix representing the heatmap, where $\mathcal{Y} = \{1, 2, \dots, m\}$ denotes the label space. For any $j, k \in \mathcal{Y}$, the $(j, k)$-th entry of HM with respect to $f_\theta$ and $\mathcal{S}$ is defined as:

$$\mathrm{HM}_{j,k} = \frac{\left|\big\{(\boldsymbol{x}, y) \in \mathcal{S} : y = j \text{ and } f_\theta\big(\mathcal{A}_{\mathrm{pgd}}(\boldsymbol{x}; y, f_\theta, \epsilon)\big) = k\big\}\right|}{\left|\big\{(\boldsymbol{x}, y) \in \mathcal{S} : y = j\big\}\right|}, \tag{8}$$

where $\mathcal{A}_{\mathrm{pgd}}$ denotes PGD attacks defined by the update rule (7). More specifically, for any $(\boldsymbol{x}, y) \in \mathcal{S}$, the PGD attack produces the corresponding adversarial example $\boldsymbol{x}' = \mathcal{A}_{\mathrm{pgd}}(\boldsymbol{x}; y, f_\theta, \epsilon)$. Then, we measure the predicted label $\hat{y} = f_\theta(\boldsymbol{x}')$. In that case, for the given training data, we could construct (*ground-truth, predicted*) label pairs, simply denoted by $\{(y, \hat{y})\}$. Afterward, we first cluster $\{(y, \hat{y})\}$ separately by the ground-truth label, e.g., the subset of ground-truth label $j$ includes all pairs such that $y = j$ (denoted by $\{(y, \hat{y})\}_j$), which corresponds to the rows of heatmaps. Further, for each subset, we group it into sub-subsets separately by the predicted labels, e.g., $\{(y, \hat{y})\}_{j,k}$ contains all pairs in $\{(y, \hat{y})\}_j$ such that $\hat{y} = k$. Consequently, the number of adversarial examples of the ground truth label $j$ is calculated as:

$$|\{(y, \hat{y})\}_j| = \left|\big\{(\boldsymbol{x}, y) \in \mathcal{S} : y = j\big\}\right|.$$

Meanwhile, the number of adversarial examples of ground truth label $j$ but predicted as label $k$ is measured as:

$$|\{(y, \hat{y})\}_{j,k}| = \left|\big\{(\boldsymbol{x}, y) \in \mathcal{S} : y = j \text{ and } f_\theta\big(\mathcal{A}_{\mathrm{pgd}}(\boldsymbol{x}; y, f_\theta, \epsilon)\big) = k\big\}\right|,$$

where $\hat{y} = f_\theta\big(\mathcal{A}_{\mathrm{pgd}}(\boldsymbol{x}; y, f_\theta, \epsilon)\big)$. Finally, we compute the $(j, k)$-th entry of the heatmap $\mathrm{HM}_{j,k}$ as the ratio of $|\{(y, \hat{y})\}_{j,k}|$ to $|\{(y, \hat{y})\}_j|$ (Equation (8)). Following the same settings, we plot the corresponding label-level variance and adversarial certainty in Figure 2. Specifically, we first measure the label-level variance of the training-time adversarial examples of the last model (Figure 1(a)) and the best model (Figure 1(c)) conditioned on the ground-truth label in Figure 2(a). Taking the ground-truth label $j$ as an example, the

label-level variance can be formulated as:

$$\text{Var}_j^{\text{(label)}} = \sqrt{\frac{1}{|\mathcal{Y}|} \sum_{k \in \mathcal{Y}} (\text{HM}_{j,k} - \overline{\text{HM}}_j)^2},$$

where $\overline{\text{HM}}_j$ averages all $\text{HM}_{j,k}$ with different $k$, and $\mathcal{Y} = \{1, 2, \cdots, m\}$ is the label space. According to Definition 1, we measure the adversarial certainty of the last and the best models, as illustrated in Figure 2(b), with respect to the predicted logits of all the adversarial examples with respect to each ground-truth label class.

## C    Proofs of Theoretical Results in Section 4

To gain a better understanding of the proposed definition of adversarial certainty, we further study its connection with robust generalization using theoretical data distributions. Following existing works Tsipras et al. (2019); Wei et al. (2023), we consider a simple binary classification task, but a further step of gradient update is considered based on our work. First, we lay out the mathematical formulations of the important concepts under the assumed setting that will be used for the proofs.

**Data Distribution.** For this binary classification task, we assume the following procedure of data generation for any example $(\boldsymbol{x}, y) \sim \mu$: The binary label $y$ is uniformly sampled, i.e., $y \overset{\text{u.a.r.}}{\sim} \{-1, +1\}$, then the robust feature $x_1 = y$ with sampling probability $p$ and $x_1 = -y$ otherwise, while the remaining non-robust features $x_2, \cdots, x_{d=1}$ are sampled i.i.d. from the Gaussian distribution $\mathcal{N}(\eta y, 1)$. Here, $p \in (\frac{1}{2}, 1)$ and $\eta < \frac{1}{2}$ is a small positive number. In general, the data distribution can be formulated as:

$$x_1 = \begin{cases} +y, & \text{w.p. } p \\ -y, & \text{w.p. } 1-p \end{cases}, \text{ and } x_2, \cdots, x_{d+1} \overset{i.i.d.}{\sim} \mathcal{N}(\eta y, 1). \tag{9}$$

**SVM Classifier.** Without bias term, an SVM classifier is used, i.e., $f(\boldsymbol{x}) = \text{sgn}(w_1 x_1 + w_2 x_2 + \cdots + w_{d+1} x_{d+1})$, where $\text{sgn}(\cdot)$ denotes the sign operator. And for brevity, we assume $w_1, w_2 \neq 0$ and $w_2 = \cdots = w_{d+1}$ as $x_2, \cdots, x_{d+1}$ are equivalent. Let $w = \frac{w_1}{w_2}$, the classifier is simplified as $f_w(\boldsymbol{x}) = \text{sgn}(x_1 + \frac{x_2 + \cdots + x_{d+1}}{w})$. And without loss of generality, since $x_2, \cdots, x_{d+1} \overset{i.i.d.}{\sim} \mathcal{N}(\eta y, 1)$ tend to share the same sign symbol with $y$, we further assume $w > 0$.

**Adversarial Distribution.** As discussed in Tsipras et al. (2019) and Ilyas et al. (2019), $x_1$ is robust to perturbation but not perfect (as $p < 1$), while $x_2, \cdots, x_{d+1}$ are useful for classification but sensitive to small perturbation. Following the setting of Tsipras et al. (2019), the non-robust features are shifted towards $-y$ by an adversarial bias distribution $\varepsilon$ for constructing adversarial examples. More specifically, the adversarial examples $\boldsymbol{x}'$ are sampled from the following adversarial distribution $\mu_{\text{adv}}(\varepsilon)$ with $\varepsilon > 0$:

$$x_1' = x_1, \text{ and } x_2', \cdots, x_{d+1}' \overset{i.i.d.}{\sim} \mathcal{N}\big((\eta - \varepsilon)y, 1\big). \tag{10}$$

Note that, in this task, no perturbation bound $\epsilon$ is involved, which is different from PGD-Attack. Instead, the distribution bias $\varepsilon$ is used to find/sample adversarial examples, which is independent of the attacker's budget. Besides, the goal of this work is to find less certain adversarial examples in the training time. As $\varepsilon$ can directly decide the distribution of adversarial examples, there is no need to vary adversarial certainty by finding a new model status.

**Robust Generalization.** Since we do not use PGD-based attacks to find adversarial examples as the empirical parts, instead of Definition 2, we utilize the corresponding version of robust generalization based on the adversarial distribution $\mu_{\text{adv}}(\varepsilon)$. Accordingly, given the model $f_w$, the clean and robust generalizations are separately denoted by $\mathcal{R}(f_w; \mu)$ and $\mathcal{R}(f_w; \mu_{\text{adv}}(\varepsilon))$, which are simply written as $\mathcal{R}_0(f_w)$ and $\mathcal{R}_\varepsilon(f_w)$ when $\mu$ and $\mu_{\text{adv}}(\varepsilon)$ are free of context:

$$\begin{aligned} \mathcal{R}_0(f_w) &= \mathbb{E}_{(\boldsymbol{x},y) \sim \mu} \mathbb{1}\big(f_w(\boldsymbol{x}) = y\big), \\ \mathcal{R}_\varepsilon(f_w) &= \mathbb{E}_{(\boldsymbol{x},y) \sim \mu_{\text{adv}}(\varepsilon)} \mathbb{1}\big(f_w(\boldsymbol{x}) = y\big). \end{aligned} \tag{11}$$

For the sake of simplicity, let robust error $\mathcal{E}_\varepsilon(f_w)$ be the robust loss for the optimization, i.e.,

$$\mathcal{E}_\varepsilon(f_w) = 1 - \mathcal{R}_\varepsilon(f_w) \tag{12}$$

The *normal distribution* $\mathcal{N}(0,1)$ is defined by the distribution function $\phi(x)$ and the probability density function $\Phi(x)$:

$$\begin{aligned}
\Phi(x) &= \int_{-\infty}^{x} \frac{1}{\sqrt{2\pi}} e^{-\frac{t^2}{2}} dt = \mathbb{P}(\mathcal{N}(0,1) < x), \\
\phi(x) &= \frac{1}{\sqrt{2\pi}} e^{-\frac{x^2}{2}} = \Phi'(x).
\end{aligned} \tag{13}$$

Recall that $w > 0$, according to Wei et al. (2023), we have

$$\mathcal{R}_0(f_w) = p\Phi(\frac{d\eta + w}{\sqrt{d}}) + (1-p)\Phi(\frac{d\eta - w}{\sqrt{d}}). \tag{14}$$

Based on the distribution of non-robust features of adversarial examples, i.e., $x_i' \sim \mathcal{N}((\eta - \varepsilon)y, 1)$, we simply replace $\eta$ with $(\eta - \varepsilon)$ in Equation 14, $\forall w > 0$, we have

$$\mathcal{R}_\varepsilon(f_w) = p\Phi(\frac{d(\eta - \varepsilon) + w}{\sqrt{d}}) + (1-p)\Phi(\frac{d(\eta - \varepsilon) - w}{\sqrt{d}}). \tag{15}$$

Consequently, we have

$$\mathcal{E}_\varepsilon(f_w) = 1 - p\Phi(\frac{d(\eta - \varepsilon) + w}{\sqrt{d}}) - (1-p)\Phi(\frac{d(\eta - \varepsilon) - w}{\sqrt{d}}). \tag{16}$$

**Adversarial Certainty.** In Section 4, we provide our definition of adversarial certainty (Definition 1) by using the empirical counterpart of adversarial distribution. However, in the theoretical part, adversarial distribution $\mu_{\text{adv}(\varepsilon)}$ is accessible. Thus, we use $\mu_{\text{adv}(\varepsilon)}$ to directly define the adversarial certainty for this binary classification task. In general, adversarial certainty measures how certain a model predicts the training-time adversarial examples, i.e., the variance of different cases of the ground-truth and the predicted labels. Based on Equation (15), all probable cases of robust generalization are:

(a) $y = +1$ and $f_w = +1$, which corresponds to robust generalization $\mathcal{R}_\varepsilon(f_w)$;

(b) $y = +1$ and $f_w = -1$, which corresponds to robust generalization $1 - \mathcal{R}_\varepsilon(f_w)$;

(c) $y = -1$ and $f_w = -1$, which corresponds to robust generalization $\mathcal{R}_\varepsilon(f_w)$;

(d) $y = -1$ and $f_w = +1$, which corresponds to robust generalization $1 - \mathcal{R}_\varepsilon(f_w)$.

As $y \overset{\text{u.a.r.}}{\sim} \{-1, +1\}$, it yields $\Pr(y = +1) = \Pr(y = -1) = \frac{1}{2}$.

For simplicity, we let $\text{AC}(f_w; \eta, \varepsilon) = \text{AC}_\varepsilon(f_w; \mu, \mu_{\text{adv}}(\varepsilon))$ in the following discussions. According to the above discussions, the adversarial certainty can be formulated as

$$\begin{aligned}
\text{AC}(f_w; \eta, \varepsilon) &= \text{Var}\Big(\mathcal{R}_\varepsilon(f_w), y, (x_1', x_2', \cdots, x_{d+1}')\Big) \\
&= \frac{1}{4}\Big[(\frac{1}{2}\mathcal{R}_\varepsilon(f_w) - \frac{1}{2})^2 + (\frac{1}{2}\mathcal{R}_\varepsilon(f_w))^2 + (\frac{1}{2}\mathcal{R}_\varepsilon(f_w) - \frac{1}{2})^2 + (\frac{1}{2}\mathcal{R}_\varepsilon(f_w))^2\Big] \\
&= \frac{1}{8}\Big[(\mathcal{R}_\varepsilon(f_w) - 1)^2 + (\mathcal{R}_\varepsilon(f_w))^2\Big] \\
&= \frac{1}{8}\Big[2\mathcal{R}_\varepsilon{}^2(f_w) - 2\mathcal{R}_\varepsilon(f_w) + 1\Big].
\end{aligned} \tag{17}$$

Now we are ready to proof Theorem 1.

### C.1 Proof of Theorem 1

*Proof of Theorem 1.* We start by showing the monotonicity of adversarial certainty with respect to $\varepsilon$.

**Monotonicity of** $\mathrm{AC}(f_w; \eta, \varepsilon)$**.** According to Equation (17), we have $\mathrm{AC}(f_w; \eta, \varepsilon) = \frac{1}{8}\Big[2\mathcal{R}_\varepsilon{}^2(f_w) - 2\mathcal{R}_\varepsilon(f_w) + 1\Big]$. Thus, the derivative to $\varepsilon$ is

$$
\begin{aligned}
\nabla_\varepsilon \mathrm{AC}(f_w; \eta, \varepsilon) &= \frac{1}{8}\Big[4\mathcal{R}_\varepsilon(f_w) \cdot \nabla_\varepsilon \mathcal{R}_\varepsilon(f_w) - 2\nabla_\varepsilon \mathcal{R}_\varepsilon(f_w)\Big] \\
&= \frac{1}{2}\Big[\mathcal{R}_\varepsilon(f_w) - \frac{1}{2}\Big] \cdot \nabla_\varepsilon \mathcal{R}_\varepsilon(f_w).
\end{aligned}
$$

In that case, to study the monotonicity of $\mathrm{AC}(f_w; \eta, \varepsilon)$, there is a need to discuss the sign of "$\mathcal{R}_\varepsilon(f_w) - \frac{1}{2}$" and "$\nabla_\varepsilon \mathcal{R}_\varepsilon(f_w)$".

$$
\nabla_\varepsilon \mathcal{R}_\varepsilon(f_w) = -\sqrt{d}p \cdot \phi\big(\frac{d(\eta - \varepsilon) + w}{\sqrt{d}}\big) - \sqrt{d}(1 - p) \cdot \phi\big(\frac{d(\eta - \varepsilon) - w}{\sqrt{d}}\big). \tag{18}
$$

As $\sqrt{d} > 0$, $0 < (1 - p) < p$, and $\phi(x) > 0$, in Equation (18), $\nabla_\varepsilon \mathcal{R}_\varepsilon(f_w) < 0$, i.e., $\mathcal{R}_\varepsilon(f_w)$ is monotonically decreasing with respect to $\varepsilon$.

According to Equation (15), we have

$$
\begin{aligned}
\mathcal{R}_\varepsilon(f_w) &= p \cdot \Phi\big(\frac{d(\eta - \varepsilon) + w}{\sqrt{d}}\big) + (1 - p) \cdot \Phi\big(\frac{d(\eta - \varepsilon) - w}{\sqrt{d}}\big) \\
&= p \cdot \int_{-\infty}^{\frac{d(\eta - \varepsilon) + w}{\sqrt{d}}} \frac{1}{\sqrt{2\pi}} e^{-\frac{t^2}{2}} dt + (1 - p) \cdot \int_{-\infty}^{\frac{d(\eta - \varepsilon) - w}{\sqrt{d}}} \frac{1}{\sqrt{2\pi}} e^{-\frac{t^2}{2}} dt.
\end{aligned}
$$

When $\varepsilon = \eta$, $d(\eta - \varepsilon) = 0$, thus

$$
\begin{aligned}
\mathcal{R}_\varepsilon(f_w) &= p \cdot \int_{-\infty}^{\frac{w}{\sqrt{d}}} \frac{1}{\sqrt{2\pi}} e^{-\frac{t^2}{2}} dt + (1 - p) \cdot \int_{-\infty}^{\frac{-w}{\sqrt{d}}} \frac{1}{\sqrt{2\pi}} e^{-\frac{t^2}{2}} dt \\
&= p \cdot \int_{-\infty}^{\frac{w}{\sqrt{d}}} \frac{1}{\sqrt{2\pi}} e^{-\frac{t^2}{2}} dt + (1 - p) - (1 - p) \cdot \int_{-\infty}^{\frac{w}{\sqrt{d}}} \frac{1}{\sqrt{2\pi}} e^{-\frac{t^2}{2}} dt \\
&= (2p - 1) \cdot \int_{-\infty}^{\frac{w}{\sqrt{d}}} \frac{1}{\sqrt{2\pi}} e^{-\frac{t^2}{2}} dt + (1 - p) \\
&> (2p - 1) \cdot \frac{1}{2} + (1 - p) \\
&= p - \frac{1}{2} + 1 - p = \frac{1}{2}.
\end{aligned}
$$

Since $\mathcal{R}_\varepsilon(f_w)$ is monotonically decreasing with respect to $\varepsilon$, when $\varepsilon \in (0, \eta]$, we have $\mathcal{R}_\varepsilon(f_w) - \frac{1}{2} > 0$. In that case,

$$
\nabla_\varepsilon \mathrm{AC}(f_w; \eta, \varepsilon) = \frac{1}{2}\Big[\mathcal{R}_\varepsilon(f_w) - \frac{1}{2}\Big] \cdot \nabla_\varepsilon \mathcal{R}_\varepsilon(f_w) < 0,
$$

that is, $\mathrm{AC}(f_w; \eta, \varepsilon)$ is monotonically decreasing with respect to $\varepsilon$ when $\varepsilon \in (0, \eta]$.

**Monotonicity of** $\mathcal{R}_{\varepsilon_{te}}(f_{\hat{w}})$**.** As aforementioned, robust error $\mathcal{E}_\varepsilon(f_w) = 1 - p\Phi(\frac{d(\eta - \varepsilon) + w}{\sqrt{d}}) - (1 - p)\Phi(\frac{d(\eta - \varepsilon) - w}{\sqrt{d}})$ is used as the robust loss to optimize $w$. In that case, the derivative of $\mathcal{E}_\varepsilon(f_w)$ to $w$ is

$$
\nabla_w \mathcal{E}_\varepsilon(f_w) = -\frac{p}{\sqrt{d}}\phi\big(\frac{d(\eta - \varepsilon) + w}{\sqrt{d}}\big) + \frac{(1 - p)}{\sqrt{d}}\phi\big(\frac{d(\eta - \varepsilon) - w}{\sqrt{d}}\big). \tag{19}
$$

Accordingly, the optimized parameters $\hat{w}$ by a step size of $\alpha > 0$ is derived as

$$
\begin{aligned}
\hat{w} &= w - \alpha \cdot \nabla_w \mathcal{E}_\varepsilon(f_w) \\
&= w + \frac{\alpha p}{\sqrt{d}} \cdot \phi\big(\frac{d(\eta - \varepsilon) + w}{\sqrt{d}}\big) - \frac{\alpha(1-p)}{\sqrt{d}} \cdot \phi\big(\frac{d(\eta - \varepsilon) - w}{\sqrt{d}}\big).
\end{aligned}
\tag{20}
$$

Following the evaluation of Tsipras et al. (2019), the non-robust features are shifted towards $-y$ to mislead $f_{\hat{w}}(\cdot)$, i.e., $\varepsilon_{te} \in [\eta, 2\eta]$, where the sampled adversarial examples follow $x_i' \sim \mathcal{N}\big((\eta - \varepsilon_{te})y, 1\big)\big|_{i=2,3,\cdots,d+1}$. Based on $\hat{w}$ and $\varepsilon_{te}$, the robust generalization is

$$
\mathcal{R}_{\varepsilon_{te}}(f_{\hat{w}}) = p \cdot \Phi\Big(\frac{d(\eta - \varepsilon_{te}) + \hat{w}}{\sqrt{d}}\Big) + (1-p) \cdot \Phi\Big(\frac{d(\eta - \varepsilon_{te}) - \hat{w}}{\sqrt{d}}\Big).
\tag{21}
$$

Accordingly, the derivative to $\hat{w}$ is

$$
\nabla_{\hat{w}} \mathcal{R}_{\varepsilon_{te}}(f_{\hat{w}}) = \frac{p}{\sqrt{d}} \cdot \phi\Big(\frac{d(\eta - \varepsilon_{te}) + \hat{w}}{\sqrt{d}}\Big) - \frac{1-p}{\sqrt{d}} \cdot \phi\Big(\frac{d(\eta - \varepsilon_{te}) - \hat{w}}{\sqrt{d}}\Big).
\tag{22}
$$

As $\varepsilon_{te} \in [\eta, 2\eta]$, $d(\eta - \varepsilon_{te}) \leq 0$. Thus, $\phi\Big(\frac{d(\eta - \varepsilon_{te}) + \hat{w}}{\sqrt{d}}\Big) \geq \phi\Big(\frac{d(\eta - \varepsilon_{te}) - \hat{w}}{\sqrt{d}}\Big)$. Consequently, $\nabla_{\hat{w}} \mathcal{R}_{\varepsilon_{te}}(f_{\hat{w}}) > 0$ when $\varepsilon_{te} \in [\eta, 2\eta]$, i.e., $\mathcal{R}_{\varepsilon_{te}}(f_{\hat{w}})$ is monotonically increasing with respect to $\hat{w}$.

**Monotonicity of $\hat{w}$.** Based on Equation (20),

$$
\begin{aligned}
\hat{w} &= w + \frac{\alpha p}{\sqrt{d}} \cdot \phi\big(\frac{d(\eta - \varepsilon) + w}{\sqrt{d}}\big) - \frac{\alpha(1-p)}{\sqrt{d}} \cdot \phi\big(\frac{d(\eta - \varepsilon) - w}{\sqrt{d}}\big) \\
&= w + \frac{\alpha p}{\sqrt{2\pi d}} \cdot e^{-\frac{\big(d(\eta-\varepsilon)+w\big)^2}{2d}} - \frac{\alpha(1-p)}{\sqrt{2\pi d}} \cdot e^{-\frac{\big(d(\eta-\varepsilon)-w\big)^2}{2d}}.
\end{aligned}
$$

In that case, the derivative of $\hat{w}$ to $\varepsilon$ is

$$
\nabla_\varepsilon \hat{w} = \frac{\alpha p}{\sqrt{2\pi d}}\big(d(\eta - \varepsilon) + w\big) \cdot e^{-\frac{(d(\eta-\varepsilon)+w)^2}{2d}} - \frac{\alpha(1-p)}{\sqrt{2\pi d}}\big(d(\eta - \varepsilon) - w\big)e^{-\frac{\big(d(\eta-\varepsilon)-w\big)^2}{2d}}.
\tag{23}
$$

When $\varepsilon \in [\eta - \frac{w}{d}, \eta]$, $d(\eta - \varepsilon) + w > 0$ and $d(\eta - \varepsilon) - w \leq 0$, thus $\nabla_\varepsilon \hat{w} > 0$, i.e., $\hat{w}$ is monotonically increasing with respect to $\varepsilon$.

**Summary.** From **Monotonicity of** $\mathrm{AC}(f_w; \eta, \varepsilon)$, we have

$$\mathrm{AC}(f_w; \eta, \varepsilon) \text{ is monotonically decreasing with respect to } \varepsilon \text{ when } \varepsilon \in (0, \eta].$$

From **Monotonicity of** $\mathcal{R}_{\varepsilon_{te}}(f_{\hat{w}})$, we have

$$\mathcal{R}_{\varepsilon_{te}}(f_{\hat{w}}) \text{ is monotonically increasing with respect to } \hat{w}.$$

From **Monotonicity of** $\hat{w}$, we have

$$\hat{w} \text{ is monotonically increasing with respect to } \varepsilon \text{ when } \varepsilon \in [\eta - \frac{w}{d}, \eta].$$

Consequently, it holds that given $f_w = \mathrm{sgn}(x_1 + \frac{x_2 + \cdots + x_{d+1}}{w})$ $(w > 0)$ and $\varepsilon \in [\eta - \frac{w}{d}, \eta]$, lower $\mathrm{AC}(f_w; \eta, \varepsilon)$, which corresponds to a larger $\varepsilon$, can yields a $f_{\hat{w}}$ with better $\mathcal{R}_{\varepsilon_{te}}(f_{\hat{w}})$ under the testing-time distribution bias $\varepsilon_{te} \in [\eta, 2\eta]$. This theoretical insight theoretically characterizes the connection between adversarial certainty and robust generalization. $\quad\square$

## C.2 Extension of Theorem 1 to $\ell_\infty$ Perturbations

Theorem 1 suggests that if we decrease the certainty of the adversarial examples sampled from $\mu_{\mathrm{adv}}(\varepsilon)$, the robustness of the SVM classifier $f_{\hat{w}}$ will increase after one-step gradient update based on the sampled adversarial examples. In this section, we generalize our theoretical analysis to the typical setting of $\ell_\infty$-norm bounded perturbations. First, we prove the following lemma to derive the adversarial data distribution with respect to worst-case $\ell_\infty$ perturbations under our problem setup.

**Lemma 2.** Consider the same data distribution and SVM classifiers as assumed in Theorem 1. For any $w > 0$ and $(\boldsymbol{x}, y)$ sampled from $\mu$, the distribution of worst-case adversarial example $(\boldsymbol{x}', y)$ under $\ell_\infty$ perturbations by using the distribution bias $\varepsilon$ is equivalent to the following adversarial data distribution:

$$x_1' = x_1 - y\varepsilon \text{ , and } x_2', \cdots, x_{d+1}' \overset{i.i.d.}{\sim} \mathcal{N}\big((\eta - \varepsilon)y, 1\big),$$

In other words, the adversarial data distribution is obtained by shifting all features of $\boldsymbol{x}$ including the robust feature $x_1$ by $y\varepsilon$. Accordingly, the robust generalization can be computed as:

$$\mathcal{R}_\varepsilon(f_w) = p \cdot \Phi\Big(\frac{d(\eta - \varepsilon) + w(1 - \varepsilon)}{\sqrt{d}}\Big) + (1 - p) \cdot \Phi\Big(\frac{d(\eta - \varepsilon) - w(1 + \varepsilon)}{\sqrt{d}}\Big). \tag{24}$$

*Proof of Lemma 2.* According to the definition of adversarial robustness in Equation (4), for any $(\boldsymbol{x}, y) \sim \mu$ and $w > 0$, the worst-case adversarial example $\boldsymbol{x}'$ under $\ell_\infty$-perturbations by using the distribution bias $\varepsilon$ is defined as:

$$\boldsymbol{x}' = \underset{\|\tilde{\boldsymbol{x}} - \boldsymbol{x}\|_\infty \leq \varepsilon}{\operatorname{argmax}} \Pr\big[f_w(\tilde{\boldsymbol{x}}) \neq y\big] = \underset{\|\tilde{\boldsymbol{x}} - \boldsymbol{x}\|_\infty \leq \varepsilon}{\operatorname{argmax}} \Pr\Big[\operatorname{sgn}\Big(\tilde{x}_1 + \frac{\tilde{x}_2 + \ldots + \tilde{x}_{d+1}}{w}\Big) \neq y\Big]. \tag{25}$$

Maximizing the objective in Equation (25) is equivalent to perturbing $\boldsymbol{x}$ in a direction such that $\tilde{x}_1 + \frac{\tilde{x}_2 + \ldots + \tilde{x}_{d+1}}{w}$ has an opposite sign to the ground-truth $y$. In the following, we are going to prove the following claim: for any $\tilde{\boldsymbol{x}}$ such that $\|\tilde{\boldsymbol{x}} - \boldsymbol{x}\|_\infty \leq \varepsilon$,

$$\Pr\Big[y \cdot \Big(\tilde{x}_1 + \frac{\tilde{x}_2 + \ldots + \tilde{x}_{d+1}}{w}\Big) < 0\Big] \leq \Pr\Big[y \cdot \Big(x_1' + \frac{x_2' + \ldots + x_{d+1}'}{w}\Big) < 0\Big], \tag{26}$$

provided that $\boldsymbol{x}'$ is defined as $x_j' = x_j - y \cdot \varepsilon$ for all $j \in \{1, \ldots, d+1\}$.

First, we have $\|\boldsymbol{x}' - \boldsymbol{x}\|_\infty = \varepsilon$ which means that $\boldsymbol{x}'$ is a feasible adversarial example. In addition, we know that for any feasible $\tilde{\boldsymbol{x}}$

$$y \cdot \Big(\tilde{x}_1 + \frac{\tilde{x}_2 + \ldots + \tilde{x}_{d+1}}{w}\Big)$$
$$\geq y \cdot \Big(x_1 + \frac{x_2 + \ldots + x_{d+1}}{w}\Big) - \Big|\tilde{x}_1 + \frac{\tilde{x}_2 + \ldots + \tilde{x}_{d+1}}{w} - \Big(x_1 + \frac{x_2 + \ldots + x_{d+1}}{w}\Big)\Big|$$
$$\geq y \cdot \Big(x_1 + \frac{x_2 + \ldots + x_{d+1}}{w}\Big) - \Big(1 + \frac{d}{w}\Big)\varepsilon$$
$$= y \cdot \Big(x_1' + \frac{x_2' + \ldots + x_{d+1}'}{w}\Big).$$

Based on the above inequalities, we immediately know that our claim specified in Equation (26) holds for any $\boldsymbol{x}$. Based on the distribution of robust feature $x_1$ and non-robust features $x_2, \ldots, x_{d+1}$ and some simple algebra to compute the robust generalization (with respect to $\boldsymbol{x}'$), we complete the proof of Lemma 2. $\square$

Now we lay out the extension of Theorem 1 to $\ell_\infty$ perturbations and its proof.

**Theorem 3.** Consider the aforementioned data distribution $\mu$ and robust classification task. Let $\varepsilon_{te} \in (\eta, 2p - 1)$ and $f_w$ be an arbitrary SVM classifier with $w > \frac{\sqrt{(d+d\eta)^2 + 16d} - (d - d\eta)}{2} > d\eta > 0$. For any

$\varepsilon \in \left(0, \min(\frac{d\eta}{w+d}, \frac{w+d\eta}{w+d} - \Delta\varepsilon)\right]$, where $\Delta\varepsilon \in (0, \frac{w+d\eta}{w+d}]$, $AC_\varepsilon(f_w; \mu, \mu_{\text{adv}}(\varepsilon))$, the adversarial certainty of $f_w$, is monotonically decreasing with respect to $\varepsilon$. Suppose we conduct one-step gradient update on $w$ using adversarial examples sampled from $\mu_{\text{adv}}(\varepsilon)$: $\hat{w} = w + \alpha \cdot \nabla_w \mathcal{R}_0(f_w; \mu_{\text{adv}}(\varepsilon))$, where $\alpha > 0$ stands for the learning rate. Then, $\mathcal{R}_0(f_{\hat{w}}; \mu_{\text{adv}}(\varepsilon_{te}))$, the robust generalization performance of $f_{\hat{w}}$, also increases as $\varepsilon$ increases.

*Proof of Theorem 3.* Similar to the proof of Theorem 1, we start by showing the monotonicity of adversarial certainty.

**Monotonicity of** $AC(f_w; \eta, \varepsilon)$**.** As defined in Equation (17), the adversarial certainty in this binary classification task is

$$AC(f_w; \eta, \varepsilon) = \frac{1}{8}\left[2\mathcal{R}_\varepsilon{}^2(f_w) - 2\mathcal{R}_\varepsilon(f_w) + 1\right].$$

And accordingly,

$$\nabla_\varepsilon AC(f_w; \eta, \varepsilon) = \frac{1}{2}\left[\mathcal{R}_\varepsilon(f_w) - \frac{1}{2}\right] \cdot \nabla_\varepsilon \mathcal{R}_\varepsilon(f_w).$$

Similarly, to study the $AC(f_w; \eta, \varepsilon)$ monotonicity, it is necessary to discuss the sign of "$\mathcal{R}_\varepsilon(f_w) - \frac{1}{2}$" and "$\nabla_\varepsilon \mathcal{R}_\varepsilon(f_w)$". According to Equation 24, the derivative of $\mathcal{R}_\varepsilon(f_w)$ to $\varepsilon$ is

$$\nabla_\varepsilon \mathcal{R}_\varepsilon(f_w) = -\frac{p(d+w)}{\sqrt{d}} \cdot \phi\Big(\frac{d(\eta - \varepsilon) + w(1 - \varepsilon)}{\sqrt{d}}\Big) \tag{27}$$
$$- \frac{(1-p)(d+w)}{\sqrt{d}} \cdot \phi\Big(\frac{d(\eta - \varepsilon) - w(1 + \varepsilon)}{\sqrt{d}}\Big).$$

As $0 < (1 - p) < \frac{1}{2} < p < 1$, $d > 0$, $w > 0$ and $\forall u, \phi(u) > 0$, it yields $\nabla_\varepsilon \mathcal{R}_\varepsilon(f_w) < 0$. That is, $\mathcal{R}_\varepsilon(f_w)$ is monotonically decreasing with respect to $\varepsilon$.

When $\varepsilon = \frac{d\eta}{w+d}$, we have $d(\eta - \varepsilon) + w(1 - \varepsilon) = w$ and $d(\eta - \varepsilon) - w(1 + \varepsilon) = -w$. Thus,

$$\begin{aligned}
\mathcal{R}_\varepsilon(f_w)\Big|_{\varepsilon=\frac{d\eta}{w+d}} &= p \cdot \Phi\Big(\frac{w}{\sqrt{d}}\Big) + (1 - p) \cdot \Phi\Big(-\frac{w}{\sqrt{d}}\Big) \\
&= p \cdot \Phi\Big(\frac{w}{\sqrt{d}}\Big) + (1 - p) - (1 - p) \cdot \Phi\Big(\frac{w}{\sqrt{d}}\Big) \\
&= (2p - 1) \cdot \Phi\Big(\frac{w}{\sqrt{d}}\Big) + (1 - p) \\
&> (2p - 1) \cdot \frac{1}{2} + (1 - p) = \frac{1}{2}.
\end{aligned}$$

In that case, $\forall \varepsilon \in (0, \frac{d\eta}{w+d}]$, it yields $\mathcal{R}_\varepsilon(f_w) - \frac{1}{2} > 0$. Consequently,

$$\nabla_\varepsilon AC(f_w; \eta, \varepsilon) = \frac{1}{2}\left[\mathcal{R}_\varepsilon(f_w) - \frac{1}{2}\right] \cdot \nabla_\varepsilon \mathcal{R}_\varepsilon(f_w) < 0,$$

that is, $AC(f_w; \eta, \varepsilon)$ is monotonically decreasing with respect to $\varepsilon$ when $\varepsilon \in (0, \frac{d\eta}{w+d}]$.

**Monotonicity of** $\mathcal{R}_{\varepsilon_{te}}(f_{\hat{w}})$**.** Similarly, the robust error $\mathcal{E}_\varepsilon(f_w) = 1 - \mathcal{R}_\varepsilon(f_w)$ is involved as the robust loss to optimize $w$. In that case, the derivative of $\mathcal{E}_\varepsilon(f_w)$ to $w$ is

$$\nabla_w \mathcal{E}_\varepsilon(f_w) = -\frac{p(1 - \varepsilon)}{\sqrt{d}} \cdot \phi\Big(\frac{d(\eta - \varepsilon) + w(1 - \varepsilon)}{\sqrt{d}}\Big) \tag{28}$$
$$+ \frac{(1-p)(1+\varepsilon)}{\sqrt{d}} \cdot \phi\Big(\frac{d(\eta - \varepsilon) - w(1 + \varepsilon)}{\sqrt{d}}\Big).$$

Accordingly, the optimized parameters $\hat{w}$ by a step size of $\alpha > 0$ is derived as

$$
\begin{aligned}
\hat{w} &= w - \alpha \cdot \nabla_w \mathcal{E}_\varepsilon(f_w) \\
&= w + \frac{\alpha p(1-\varepsilon)}{\sqrt{d}} \cdot \phi\Big(\frac{d(\eta-\varepsilon) + w(1-\varepsilon)}{\sqrt{d}}\Big) - \frac{\alpha(1-p)(1+\varepsilon)}{\sqrt{d}} \cdot \phi\Big(\frac{d(\eta-\varepsilon) - w(1+\varepsilon)}{\sqrt{d}}\Big).
\end{aligned}
\tag{29}
$$

Based on $\hat{w}$ and $\varepsilon_{te}$, the robust generalization is

$$
\mathcal{R}_{\varepsilon_{te}}(f_{\hat{w}}) = p \cdot \Phi\Big(\frac{d(\eta-\varepsilon_{te}) + \hat{w}(1-\varepsilon_{te})}{\sqrt{d}}\Big) + (1-p) \cdot \Phi\Big(\frac{d(\eta-\varepsilon_{te}) - \hat{w}(1+\varepsilon_{te})}{\sqrt{d}}\Big).
\tag{30}
$$

Accordingly, the derivative to $\hat{w}$ is

$$
\begin{aligned}
\nabla_{\hat{w}} \mathcal{R}_{\varepsilon_{te}}(f_{\hat{w}}) =\ &\frac{p(1-\varepsilon_{te})}{\sqrt{d}} \cdot \phi\Big(\frac{d(\eta-\varepsilon_{te}) + \hat{w}(1-\varepsilon_{te})}{\sqrt{d}}\Big) \\
&- \frac{(1-p)(1+\varepsilon_{te})}{\sqrt{d}} \cdot \phi\Big(\frac{d(\eta-\varepsilon_{te}) - \hat{w}(1+\varepsilon_{te})}{\sqrt{d}}\Big).
\end{aligned}
\tag{31}
$$

As $\eta \leq \varepsilon_{te} \leq (2p-1)$, it yields $0 < \frac{(1-p)(1+\varepsilon_{te})}{\sqrt{d}} < \frac{p(1-\varepsilon_{te})}{\sqrt{d}}$, and $0 < \phi\Big(\frac{d(\eta-\varepsilon_{te}) - \hat{w}(1+\varepsilon_{te})}{\sqrt{d}}\Big) < \phi\Big(\frac{d(\eta-\varepsilon_{te}) + \hat{w}(1-\varepsilon_{te})}{\sqrt{d}}\Big)$, thus $\nabla_{\hat{w}} \mathcal{R}_{\varepsilon_{te}}(f_{\hat{w}}) > 0$. That is, $\mathcal{R}_{\varepsilon_{te}}(f_{\hat{w}})$ is monotonically increasing with respect to $\hat{w}$.

**Monotonicity of $\hat{w}$.** Based on Equation 29,

$$
\hat{w} = w + \frac{\alpha p(1-\varepsilon)}{\sqrt{d}} \cdot \phi\Big(\frac{d(\eta-\varepsilon) + w(1-\varepsilon)}{\sqrt{d}}\Big) - \frac{\alpha(1-p)(1+\varepsilon)}{\sqrt{d}} \cdot \phi\Big(\frac{d(\eta-\varepsilon) - w(1+\varepsilon)}{\sqrt{d}}\Big).
$$

In that case, the derivative of $\hat{w}$ to $\varepsilon$ is

$$
\begin{aligned}
\nabla_\varepsilon \hat{w} =\ &\frac{\alpha p}{\sqrt{d}}\Big[-\phi\Big(\frac{d(\eta-\varepsilon)+w(1-\varepsilon)}{\sqrt{d}}\Big) + (1-\varepsilon)\cdot\nabla_\varepsilon\phi\Big(\frac{d(\eta-\varepsilon)+w(1-\varepsilon)}{\sqrt{d}}\Big)\Big] \\
&- \frac{\alpha(1-p)}{\sqrt{d}}\Big[\phi\Big(\frac{d(\eta-\varepsilon)-w(1+\varepsilon)}{\sqrt{d}}\Big) + (1+\varepsilon)\cdot\nabla_\varepsilon\phi\Big(\frac{d(\eta-\varepsilon)-w(1+\varepsilon)}{\sqrt{d}}\Big)\Big] \\
=\ &\frac{\alpha p}{\sqrt{d}}\Big[-1 + \frac{(1-\varepsilon)(d+w)\big(d(\eta-\varepsilon)+w(1-\varepsilon)\big)}{d}\Big]\cdot\phi\Big(\frac{d(\eta-\varepsilon)+w(1-\varepsilon)}{\sqrt{d}}\Big) \\
&- \frac{\alpha(1-p)}{\sqrt{d}}\Big[1 + \frac{(1+\varepsilon)(d+w)\big(d(\eta-\varepsilon)-w(1+\varepsilon)\big)}{d}\Big]\cdot\phi\Big(\frac{d(\eta-\varepsilon)-w(1+\varepsilon)}{\sqrt{d}}\Big) \\
=\ &\frac{\alpha p}{d\sqrt{d}}\Big[\big((w+d)\varepsilon - (w+d)\big)\big((w+d)\varepsilon - (w+d\eta)\big) - d\Big]\cdot\phi\Big(\frac{d(\eta-\varepsilon)+w(1-\varepsilon)}{\sqrt{d}}\Big) \\
&+ \frac{\alpha(1-p)}{d\sqrt{d}}\Big[\big((w+d)\varepsilon + (w+d)\big)\big((w+d)\varepsilon + (w-d\eta)\big) - d\Big]\cdot\phi\Big(\frac{d(\eta-\varepsilon)-w(1+\varepsilon)}{\sqrt{d}}\Big).
\end{aligned}
\tag{32}
$$

As $d\eta < \frac{\sqrt{(d+d\eta)^2+16d}-(d-d\eta)}{2} < w$, it yields $\big((w+d)\varepsilon + (w+d)\big)\big((w+d)\varepsilon + (w-d\eta)\big) - d > 0$. As $0 < \frac{w+d\eta}{w+d} < \frac{w+d}{w+d}$ and $\big((w+d)\varepsilon - (w+d)\big)\big((w+d)\varepsilon - (w+d\eta)\big)\big|_{\varepsilon=0} > d$, it yields $\exists \Delta\varepsilon \in (0, \frac{w+d\eta}{w+d}]$, such that $\big((w+d)\varepsilon - (w+d)\big)\big((w+d)\varepsilon - (w+d\eta)\big)\big|_{\varepsilon=\frac{w+d\eta}{w+d}-\Delta\varepsilon} > d$. In that case, it holds that $\forall \varepsilon \in (0, \frac{w+d\eta}{w+d} - \Delta\varepsilon]$, $\nabla_\varepsilon \hat{w} > 0$, that is, $\hat{w}$ is monotonically increasing with respect to $\varepsilon$ when $\varepsilon \in (0, \frac{w+d\eta}{w+d} - \Delta\varepsilon]$.

**Summary.** From **Monotonicity of** $\mathrm{AC}(f_w; \eta, \varepsilon)$, we have

$$
\mathrm{AC}(f_w; \eta, \varepsilon) \text{ is monotonically decreasing with respect to } \varepsilon \text{ when } \varepsilon \in (0, \frac{d\eta}{w+d}].
$$

From **Monotonicity of** $\mathcal{R}_{\varepsilon_{te}}(f_{\hat{w}})$, we have

$$
\mathcal{R}_{\varepsilon_{te}}(f_{\hat{w}}) \text{ is monotonically increasing with respect to } \hat{w}.
$$

From **Monotonicity of** $\hat{w}$, we have

$$\hat{w} \text{ is monotonically increasing with respect to } \varepsilon \text{ when } \varepsilon \in (0, \frac{w+d\eta}{w+d} - \Delta\varepsilon].$$

Consequently, it holds that given $f_w = \text{sgn}(x_1 + \frac{x2+\cdots+x_{d+1}}{w})$ $(0 < d\eta < \frac{\sqrt{(d+d\eta)^2+16d}-(d-d\eta)}{2} < w)$ and $\varepsilon \in (0, \min(\frac{d\eta}{w+d}, \frac{w+d\eta}{w+d} - \Delta\varepsilon)]$, where $\Delta\varepsilon \in (0, \frac{w+d\eta}{w+d}]$, lower $\text{AC}(f_w; \eta, \varepsilon)$, which corresponds to a larger $\varepsilon$, can yields a $f_{\hat{w}}$ with better $\mathcal{R}_{\varepsilon_{te}}(f_{\hat{w}})$ under the testing-time distribution bias $\varepsilon_{te} \in [\eta, 2p-1]$. $\square$

## D  Significance Test for the Improvements of DAC on AWP and Consistency

As discussed in Section 6.2, our DAC method can only bring slight improvements in robust generalization for AWP and Consistency. Although our repeated trials have suggested that the improvements are the consequence of DAC (see Table 6), it is helpful to provide some statistical support. Therefore, in this section, we conduct a $t$-test to measure the statistical significance of our DAC method.[1] Specifically, we first make a null hypothesis, i.e.,

$$\text{H}_0 : \textit{Our DAC method does not improve the robust generalization of AWP and Consistency.}$$

We then collect the robust generalization under AutoAttack without and with DAC from Table 6, which are separately denoted as two samples $X_1$ and $X_2$. This decision is because AutoAttack is more powerful and comprehensive than other adversarial attacks used in our evaluation, and AutoAttack is now the default metric for the leaderboard of adversarial defenses.[2] In that case, the null hyperthesis $\text{H}_0$ can be informally understood as $X_1 \geq X_2$. Next, we calculate the mean of $X_1$ and $X_2$, which can be formulated as:

$$\bar{X}_1 = \frac{1}{n_1} \sum_{i=1}^{n_1} X_{1i}, \text{ and } \bar{X}_2 = \frac{1}{n_2} \sum_{i=1}^{n_2} X_{2i},$$

where $n_1$ and $n_2$ are the size of $X_1$ and $X_2$ in this case. Subsequently, the standard deviation of $X_1$ and $X_2$ can be calculated as:

$$s_1 = \sqrt{\frac{1}{n_1 - 1} \sum_{i=1}^{n_1} \left(X_{1i} - \bar{X}_1\right)^2}, \text{ and } s_2 = \sqrt{\frac{1}{n_2 - 1} \sum_{i=1}^{n_2} \left(X_{2i} - \bar{X}_2\right)^2}.$$

Accordingly, the pooled standard deviation of the two samples is represented by $s_1$ and $s_2$, i.e.,

$$s_p = \sqrt{\frac{(n_1 - 1)s_1^2 + (n_2 - 1)s_2^2}{n_1 + n_2 - 2}},$$

where $n_1 + n_2 - 2$ is the total number of degrees of freedom. Given $\bar{X}_1$, $\bar{X}_2$ and $s_p$, we have t-statistic

$$t = \left| \frac{\bar{X}_1 - \bar{X}_2}{s_p \sqrt{\frac{1}{n_1} + \frac{1}{n_2}}} \right|.$$

In this case, we have the t-statistic $t = 2.141$ and total number of degrees of freedom $n_1 + n_2 - 2 = 22$. By comparing to the $t$-Table, our t-statistic $t$ is larger than the element of $t_{.975} = 2.074$, i.e., we have $> 95\%$ confidence to reject the null hypothesis $H_0$.[3] In other words, our DAC method can bring statistically significant improvements to AWP and Consistency.

## E  Complete Results

Due to the space limit, we present the full results of Table 2 and Table 3 in Table 7 and Table 8, respectively.

---

[1] https://en.wikipedia.org/wiki/Student%27s_t-test
[2] https://robustbench.github.io/
[3] https://www.sjsu.edu/faculty/gerstman/StatPrimer/t-table.pdf

Table 7: Testing-time adversarial robustness (%) of AT with/without DAC/DAC_Reg under $\ell_\infty$ perturabtions across different model architectures and benchmark datasets.

| Dataset | Architecture | Method | Clean | PGD-20 | PGD-100 | CW$_\infty$ | AutoAttack |
|---|---|---|---|---|---|---|---|
| SVHN | PRN18 | AT | 89.63 (88.64) | 42.25 (51.00) | 41.37 (50.30) | 42.84 (48.19) | 39.52 (46.02) |
| | | + DAC | 90.58 (89.63) | **45.86** (**54.42**) | **43.92** (**53.78**) | **43.75** (**50.15**) | 40.68 (**48.23**) |
| | | + DAC_Reg | **90.65** (**90.21**) | 45.39 (53.06) | 43.77 (52.28) | 43.66 (49.64) | **41.10** (47.39) |
| | WRN34 | AT | 91.51 (89.72) | 46.81 (53.43) | 44.94 (52.77) | 45.76 (50.43) | 41.71 (49.50) |
| | | + DAC | 91.26 (91.83) | 60.42 (**67.95**) | 56.71 (**64.85**) | 56.98 (**65.09**) | 42.33 (**50.42**) |
| | | + DAC_Reg | **91.76** (**92.13**) | **62.19** (65.96) | **59.54** (63.68) | **60.05** (63.87) | **42.46** (49.95) |
| CIFAR-10 | PRN18 | AT | 82.88 (82.68) | 41.51 (49.23) | 40.96 (48.92) | 41.61 (48.07) | 39.66 (45.71) |
| | | + DAC | **84.64** (**83.55**) | **45.55** (**52.20**) | **44.94** (**51.87**) | **44.55** (**50.05**) | **42.78** (**48.20**) |
| | | + DAC_Reg | 83.78 (83.54) | 45.39 (50.86) | 44.87 (50.49) | 44.18 (48.96) | 42.41 (47.02) |
| | WRN34 | AT | 86.47 (**85.86**) | 47.25 (55.31) | 46.73 (55.00) | 47.85 (54.04) | 45.84 (51.94) |
| | | + DAC | **86.48** (85.10) | **52.02** (**57.93**) | **51.69** (**57.68**) | **51.51** (**54.98**) | **49.75** (**53.33**) |
| | | + DAC_Reg | 85.69 (76.89) | 48.81 (48.91) | 47.54 (48.86) | 47.55 (45.98) | 44.24 (44.99) |
| CIFAR-100 | PRN18 | AT | 54.58 (53.64) | 20.29 (27.80) | 20.00 (27.66) | 20.18 (25.40) | 18.52 (23.45) |
| | | + DAC | **54.85** (**55.01**) | **22.46** (27.73) | **22.19** (27.48) | **21.11** (25.37) | 19.09 (**23.95**) |
| | | + DAC_Reg | 54.67 (53.11) | 21.78 (**28.86**) | 21.50 (**28.70**) | 20.56 (**26.00**) | **19.29** (23.40) |
| | WRN34 | AT | 57.23 (54.45) | 25.64 (30.30) | 25.38 (29.97) | 24.09 (27.57) | 22.76 (25.46) |
| | | + DAC | **58.15** (58.04) | **26.08** (**31.55**) | **25.89** (**31.43**) | **24.77** (**29.19**) | **23.66** (**27.08**) |
| | | + DAC_Reg | 57.57 (**58.34**) | 24.46 (30.97) | 24.13 (30.89) | 24.04 (28.92) | 22.68 (26.71) |

Table 8: Testing-time adversarial robustness (%) of AT, TRADES and MART with/without DAC on SVHN, CIFAR-10 and CIFAR-100 under $\ell_\infty$ perturbations.

| Dataset | Method | Clean | PGD-20 | PGD-100 | CW$_\infty$ | AutoAttack |
|---|---|---|---|---|---|---|
| SVHN | AT | 89.63 (88.64) | 42.25 (51.00) | 41.37 (50.30) | 42.84 (48.19) | 39.52 (46.02) |
| | + DAC | 90.58 (89.63) | **45.86** (**54.42**) | **43.92** (**53.78**) | **43.75** (**50.15**) | 40.68 (**48.23**) |
| | + DAC_Reg | **90.65** (**90.21**) | 45.39 (53.06) | 43.77 (52.28) | 43.66 (49.64) | **41.10** (47.39) |
| | TRADES | 89.12 (87.75) | 51.50 (55.19) | 50.69 (54.50) | 45.50 (50.32) | 45.02 (48.69) |
| | + DAC | **90.24** (89.59) | **52.24** (**57.09**) | **51.14** (**56.39**) | **46.34** (**52.22**) | **46.20** (**50.52**) |
| | + DAC_Reg | 90.03 (**89.75**) | 51.78 (56.10) | 50.92 (54.83) | 45.86 (51.35) | 45.30 (49.06) |
| | MART | 89.68 (84.48) | 49.07 (52.30) | 48.30 (52.22) | 45.48 (48.04) | 44.54 (47.38) |
| | + DAC | 88.90 (84.64) | **51.04** (**53.64**) | **50.91** (**52.70**) | **46.94** (**49.96**) | **46.18** (**48.50**) |
| | + DAC_Reg | **90.18** (**88.47**) | 50.94 (52.94) | 49.87 (52.46) | 46.32 (49.18) | 45.86 (47.73) |
| CIFAR-10 | AT | 82.88 (82.68) | 41.51 (49.23) | 40.96 (48.92) | 41.61 (48.07) | 39.66 (45.71) |
| | + DAC | **84.64** (**83.55**) | **45.55** (**52.20**) | **44.94** (**51.87**) | **44.55** (**50.05**) | **42.78** (**48.20**) |
| | + DAC_Reg | 83.78 (83.54) | 45.39 (50.86) | 44.87 (50.49) | 44.18 (48.96) | 42.41 (47.02) |
| | TRADES | 82.10 (81.33) | 47.44 (51.65) | 46.95 (51.42) | 46.64 (49.18) | 44.99 (48.06) |
| | + DAC | **83.18** (**82.80**) | **49.32** (52.90) | **48.81** (**52.67**) | **48.30** (**50.11**) | **46.40** (**48.96**) |
| | + DAC_Reg | 82.97 (82.37) | 47.87 (**53.33**) | 47.33 (51.88) | 46.80 (49.68) | 45.07 (48.70) |
| | MART | 80.85 (78.27) | 50.23 (52.28) | 49.71 (52.13) | 46.88 (47.83) | 44.68 (46.01) |
| | + DAC | **81.12** (**79.37**) | **52.38** (53.25) | **52.04** (53.14) | **48.97** (49.25) | **47.24** (**47.69**) |
| | + DAC_Reg | 80.81 (79.07) | 50.35 (52.71) | 50.06 (52.54) | 47.50 (**49.32**) | 45.72 (47.19) |
| CIFAR-100 | AT | 54.58 (53.64) | 20.29 (27.80) | 20.00 (27.66) | 20.18 (25.40) | 18.52 (23.45) |
| | + DAC | **54.85** (**55.01**) | **22.46** (27.73) | **22.19** (27.48) | **21.11** (25.37) | 19.09 (**23.95**) |
| | + DAC_Reg | 54.67 (53.11) | 21.78 (**28.86**) | 21.50 (**28.70**) | 20.56 (**26.00**) | **19.29** (23.40) |
| | TRADES | 55.40 (53.98) | 22.40 (28.31) | 22.32 (28.18) | 21.42 (25.82) | 20.55 (24.29) |
| | + DAC | **56.66** (**54.67**) | **25.54** (**29.56**) | **25.43** (**29.35**) | **23.32** (**25.92**) | **22.35** (**24.87**) |
| | + DAC_Reg | 54.36 (53.86) | 23.16 (27.96) | 23.13 (27.88) | 22.34 (24.35) | 21.08 (23.43) |
| | MART | 55.73 (52.48) | 24.18 (27.17) | 24.00 (27.12) | 22.41 (24.89) | 21.56 (23.06) |
| | + DAC | **55.94** (**54.23**) | **24.81** (**28.23**) | **24.65** (**28.13**) | **23.53** (**25.19**) | **22.06** (**24.00**) |
| | + DAC_Reg | 55.52 (52.77) | 24.33 (27.40) | 24.21 (23.41) | 22.83 (25.04) | 21.73 (23.28) |

## F  Additional Evaluations

In this section, we provide additional experimental results better to comprehend our work, including the influence of $\epsilon$ on DAC and training efficiency.

### F.1  Influence of $\epsilon$ on DAC

Table 9: Testing-time robust generalization (RG%) on the last and best epochs, where the training-time $\epsilon$ is set to $8/255$ while the testing one varies in $\{4/255, 8/255, 12/255\}$.

| $\epsilon_{\mathbf{tr}} = \mathbf{8/255}$ | $\epsilon_{\mathbf{te}} = \mathbf{4/255}$ | $\epsilon_{\mathbf{te}} = \mathbf{8/255}$ | $\epsilon_{\mathbf{te}} = \mathbf{12/255}$ |
|---|---|---|---|
| $\mathbf{RG}_{\text{last}}/\mathbf{RG}_{\text{best}}$ | 66.03/70.09 | 45.55/52.20 | 29.91/34.88 |

Table 10: Testing-time robust generalization (RG%) on the last and best epochs, where the testing-time $\epsilon$ is set to $8/255$ while the training one varies in $\{4/255, 8/255, 12/255\}$.

| $\epsilon_{\mathbf{te}} = \mathbf{8/255}$ | $\epsilon_{\mathbf{tr}} = \mathbf{4/255}$ | $\epsilon_{\mathbf{tr}} = \mathbf{8/255}$ | $\epsilon_{\mathbf{tr}} = \mathbf{12/255}$ |
|---|---|---|---|
| $\mathbf{RG}_{\text{last}}/\mathbf{RG}_{\text{best}}$ | 41.49/45.51 | 45.55/52.20 | 47.96/53.32 |

In the previous evaluation, our work focuses on the setting of $\epsilon = 8/255$ in both the training and testing time for $\ell_\infty$-norm perturbations, which is widely used in this field as $\epsilon$ is usually associated with the assumption of the adversarial strength. However, it is interesting to investigate the influence of $\epsilon$ on our DAC method. Thus, we separately fixed the training- and testing-time $\epsilon$ to $8/255$, and vary the other one in $\{4/255, 8/255, 12/255\}$, as shown in Tables 9 and 10. Specifically, Table 9 depicts that the model of a fixed training-time $\epsilon_{tr}$ is more vulnerable to a larger testing-time $\epsilon_{te}$, on both the last and best epochs. On the other hand, Table 10, shows that a stronger adversarial ability in the training time can derive a better robustness when facing the same testing-time $\epsilon_{te}$. Consequently, these results suggest that model robustness will benefit from a larger $\epsilon_{tr}$ and a smaller $\epsilon_{te}$.

### F.2  Training Efficiency

Table 11: Testing-time robust generalization (RG %) on the last and best epochs, where $\text{DAC}_t$ indicates that decreasing adversarial certainty starts from the $t$-th epoch.

| RG (%) | Last | Best |
|---|---|---|
| $\text{DAC}_1$ | 45.55 | 52.20 |
| $\text{DAC}_{101}$ | 45.84 | 50.49 |
| $\text{DAC}_{151}$ | 45.68 | 49.85 |

In Section 6.4, we proposed DAC_Reg to reduce the training cost. However, it modifies the optimization flow. Therefore, to enhance DAC's efficiency, we conduct an additional attempt by decreasing adversarial certainty only when robust overfitting occurs. In our evaluation, we start the DAC method from the 101-th and 151-th epochs, respectively, corresponding to the first and second time of learning rate decay. From Table F.2, we can see that an earlier application of DAC brings better "Best" robustness, while the "Last" one is comparable. Consequently, improving training efficiency by reducing the epochs that consider DAC can only derive limited performance promotion. In other words, the potential direction of developing efficient methods might be regularizing some specific insights in a loss term, such as the design of DAC_Reg. Still, the effort to guarantee stability is necessary.

**Potential Limitation.** Table 11 depicts that applying DAC earlier can achieve better robustness. Thus, $\text{DAC}_{101}$ and $\text{DAC}_{151}$, decreasing adversarial certainty when robust overfitting happens, cannot achieve comparable robustness on the best epoch with $\text{DAC}_1$ that works from scratch. That is, our method requires a warm-up instead of an immediate functionality. Consequently, our method might not be suitable for the real-time scenario, where DAC is deployed only when robust overfitting happens, which limits the flexibility of our method.

