# OpenReview forum: "Generating Less Certain Adversarial Examples Improves Robust Generalization"
_TMLR — Accepted by TMLR_

### Review · Reviewer_5svJ · 2024-08-30

**Summary Of Contributions:**

Robust overfitting is widely observed in existing adversarially trained models, where robust generalization keeps decreasing after an early time. This paper claims this is because those models are too confident classifying adversarial examples. This paper introduces ‘adversarial certainty’, hypothesizes generating less certain adversarial examples improves robust generalization, and backs this up with theory and numerics. Theoretically, connection between adversarial certainty and robust generalization are studied through synthetic distributions. Based on above hypothesis, the proposed method, used as additional updates (2 gradient updates in eqn. 3) after existing robust methods, first explicitly decreases adversarial certainty (DAC) by gradient descent (GD), then use new uncertain adversarial examples to optimize robustness by GD. This approach mitigates adversarial overconfidence. More analysis shows the proposed adversarial certainty is equivalent to a certain regularization term. Numerical results on image data show proposed method combined after existing methods produces decently more robust models than existing adversarial training methods alone.

**Audience:**

Yes

**Claims And Evidence:**

Yes

**Requested Changes:**

1. [unclear grammar] The meaning of ‘generating less certain adversarial examples’ in [1) abstract, line 5; 2) P2, line 4; 3) P2, line 15] is unclear and confusing. This should be replaced by more sensible phrases such as ‘decreasing adversarial certainty’, or more detailed words such as [P2, line 5]’s next sentence.
2. [P7, correlation analysis] ‘Does decreasing adversarial certainty always induce better robust generalization?’ is a crucial question, but currently only addressed mostly by heuristics. The partially theoretical claim ‘this result suggests that…’ is not sufficiently rigorous and sensible. If possible, more rigorous details are desirable.

**Strengths And Weaknesses:**

Strengths:
1. The proposed idea is sensible. The method, being a 2 line gradient add-on to existing adversarial training method, is simple and easy to implement.
2. The idea and method is decently covered by theoretical analysis. Theoretically, authors first prove in the special case of linear SVM classifier, as adversarial perturbation increases, adversarial certainty decreases and (after gradient update) robust generalization increases monotonically. Authors then show 1) in terms of numerical correlation, (in Fig. 2c) adversarial certainty is negatively correlated with robust generalization when step size is sufficiently small, 2) in terms of theory, *possibly* when in feasible search region with small step size, decreasing adversarial certainty induces better robust generalization; this was based on observation but not proven.
3. Numerical results show this simple add-on approach usually achieves some improvement in robustness, when added onto sufficiently many existing adversarial training methods, or added on top of other robust add-ons.

Weaknesses:
1.  For the main hypothesis that less adversarial certainty means better robustness, the theoretical analysis is not complete proof, but only supporting evidence (see my strength 2). There are very significant gaps from theoretical analysis to completely proving the hypothesis. Even the limited claim in [strength 2.2)], as an attempt to bridge above gap, is only observed but not proven.
2. Some short summaries in early pages are unclear and confusing. One example is requested changes 1.

---

> ### Author Response · Authors · 2024-09-30
> **Response to Reviewer 5svJ**
>
> We appreciate the reviewer's positive feedback on our proposed idea, theoretical analysis, and numerical results. Below, we provide our responses to the required changes.
>
> > [Unclear grammar]
> > The meaning of ‘generating less certain adversarial examples’ in [1) abstract, line 5; 2) P2, line 4; 3) P2, line 15] is unclear and confusing. This should be replaced by more sensible phrases such as ‘decreasing adversarial certainty’, or more detailed words such as [P2, line 5]’s next sentence.
>
> In the revision, we will provide clearer and more detailed clarifications by adaptively replacing "generating less certain adversarial examples" with "generating adversarial examples after the optimization of decreasing adversarial certainty".
>
> > [Correlation analysis]
> > "Does decreasing adversarial certainty always induce better robust generalization?" is a crucial question, but currently only addressed mostly by heuristics. The partially theoretical claim "this result suggests that…" is not sufficiently rigorous and sensible. If possible, more rigorous details are desirable.
>
> Table 4: Training-time adversarial certainty (AC) and testing-time robust generalization (RG %) of one-epoch optimization, where the adversarial certainty is decreased by $t$ times with a step size of $0.7/t$.
> |$t$|1|2|3|4|5|
> |:-|:-:|:-:|:-:|:-:|:-:|
> |AC|1.1834|1.0324|0.8824|0.8015|0.6879|
> |RG|44.09|44.03|44.51|45.38|45.56|
> ||6|7|8|9|10|
> |AC|0.6782|0.5958|0.5925|0.5249|0.5138|
> |RG|45.61|45.95|45.87|44.42|44.40|
>
> Table 5: Training-time adversarial certainty (AC) and testing-time robust generalization (RG %) of one-epoch optimization starting from the best model with various step sizes from $0.1$ to $2.0$.
> |Step Size|0.1|0.2|0.3|0.4|0.5|
> |:-|:-:|:-:|:-:|:-:|:-:|
> |AC|1.8587|1.5283|1.3187|1.1637|1.0402|
> |RG|48.25|49.55|49.53|50.08|50.31|
> ||0.6|0.7|0.8|0.9|1.0|
> |AC|0.9368|0.8452|0.7647|0.6863|0.6175|
> |RG|50.15|50.94|50.66|50.18|50.07|
> ||1.1|1.2|1.3|1.4|1.5|
> |AC|0.5461|0.4855|0.4314|0.3706|0.2882|
> |RG|49.72|49.94|50.33|49.81|50.12|
> ||1.6|1.7|1.8|1.9|2.0|
> |AC|0.2440|0.1725|0.1354|0.0948|0.0910|
> |RG|50.37|49.47|49.25|49.85|50.06|
>
> Our correlation analysis (Section 5 and Figure 2\(c\)) provided a negative answer by conducting one-epoch optimization with different step sizes on the last model of AT, i.e., the model of Figures 1\(a\) and 1\(b\). Here, we demonstrate that our finding is indeed from the changes of adversarial certainty, but not derived from the one-epoch setting. Specifically, we separate one step of decreasing adversarial certainty with size $0.7$ into $t$ steps, i.e., each step corresponds to the size of $0.7/t$, where $t=\\{1,2,3,4,5,6,7,8,9,10\\}$. The results of training-time adversarial certainty (AC) and testing-time robust generalization (RG %) are depicted in Table 4, where we can observe a consistent pattern with Figure 2\(c\) that RG first gains more improvements but then less with the decrease of AC.
>
> Moreover, we apply the one-epoch optimization on the best AT model that corresponds to Figures 1\(c\) and 1\(d\) to derive more observations. From Table 5, we can find that the changes of RG with the increase of AC depict a different pattern from Figure 2\(c\). First, the improvements on RG are obviously slighter than those on the last model. Besides, there is no trend of RG decreasing even if the step size arrives at $2.0$. These results suggest that the feasible region of the best model is larger than that of the last model which suffers more severe robust overfitting. Consequently, when applying our DAC method, an overconfident model is supposed to select the optimization step size carefully; otherwise, the selection could be more ambitious.
>
> We hope our complementary results can provide more empirical evidence of our correlation analysis.

---

> > ### Comment · Reviewer_5svJ · 2024-10-02
> > **Follow-up Response to Author**
> >
> > Thank you very much for the response and updates. My requested change 2 still has some gaps, but my other concerns are mostly addressed.

---

> > > ### Author Response · Authors · 2024-10-02
> > > **Our Appreciation to Reviewer 5svj**
> > >
> > > We thank the reviewer's feedback and are happy that our response can solve the most concerns. We are glad to hear any further suggestions for shrinking the gap of the requested change 2, and include the corresponding solutions in the revision if possible.

---

### Review · Reviewer_XCB7 · 2024-09-14

**Summary Of Contributions:**

This paper addresses a key issue in adversarial training—robust overfitting—and introduces a new concept, "adversarial certainty," to quantify the model's confidence in its predictions. By reducing the certainty in adversarial training, i.e., increasing the variability of model outputs, the paper aims to enhance the model's robust generalization against adversarial samples. Through extensive experiments and theoretical analysis, the paper demonstrates that the proposed method effectively mitigates robust overfitting in adversarial training and showcases performance improvements on multiple standard datasets.

**Audience:**

Yes

**Broader Impact Concerns:**

No ethical implications.

**Claims And Evidence:**

Yes

**Requested Changes:**

1. It is suggested to either extend the method to additional data types and tasks (e.g., natural language processing or audio tasks) to further validate its generalization capability, or if adding experiments is difficult, provide relevant discussion on how the method might generalize to other domains.

2. While improving training efficiency through experiments would be beneficial, if conducting additional experiments is challenging, discussing potential optimizations or alternatives could provide valuable insights. This can include suggestions for future work on how the method could be made more computationally efficient.

3. If possible, include detailed case studies or additional experiments where the method may not perform optimally. However, if resource constraints prevent additional experimentation, offering a discussion on potential limitations based on existing results would also help to clarify the boundaries of the method's effectiveness.

**Strengths And Weaknesses:**

Strengths
- Theoretical Contribution: The introduction of adversarial certainty provides a new perspective for understanding model behavior in adversarial training, supported by theoretical analysis that suggests reducing certainty can enhance model robustness.
- Experimental Validation: Extensive experiments across various standard datasets validate that reducing adversarial certainty improves robust generalization capabilities, particularly highlighted through comparisons with existing technologies.
- Practicality: The proposed method can be directly applied to existing adversarial training frameworks, showing good practicality and potential for widespread adoption.

Weaknesses
- Increased Complexity: Although theoretically effective, this method may increase the complexity and computational demands of the training process, particularly in large-scale datasets and models, potentially significantly extending training times.
- Generalization Verification: The paper focuses on image recognition tasks, and the effectiveness of this approach for other types of tasks (such as natural language processing or audio processing) remains to be further researched and verified.
- Parameter Sensitivity: The model and methods discussed may be sensitive to hyperparameter choices, such as step size and regularization coefficients, which could impact final performance and stability.

---

> ### Author Response · Authors · 2024-09-30
> **Response to Reviewer XCB7**
>
> We appreciate the reviewer's positive feedback on our theoretical contribution, experimental validation, and practicality. Below, we provide our responses to the required changes.
>
> > [Generalization to NLP]
> > It is suggested to either extend the method to additional data types and tasks (e.g., natural language processing or audio tasks) to further validate its generalization capability, or if adding experiments is difficult, provide relevant discussion on how the method might generalize to other domains.
>
> Natural language processing (NLP) is a different topic from computer vision (CV), as it focuses on the discrete space while that of CV is continuous. Thus, directly transferring our method to the NLP domain is non-trivial. According to the pipeline of vanilla adversarial training in NLP [1], the generation of adversarial examples is to substitute some selected words. Such a generation scheme is different from that of CV and does not support the gradient-based optimization of decreasing adversarial certainty. In that case, we would expect that an additional objective function is designed to measure how certain an NLP model is on its generated sentences, which could be the metric for selecting and substituting words. Nevertheless, we regard the exploration of DAC in other domains such as NLP as interesting future work but beyond the scope of our work. We believe our proposed concept of adversarial certainty is important for the development of robust computer vision systems.
>
> [1] Reevaluating Adversarial Examples in Natural Language, Morris et al.
>
> > [Training efficiency]
> > While improving training efficiency through experiments would be beneficial, if conducting additional experiments is challenging, discussing potential optimizations or alternatives could provide valuable insights. This can include suggestions for future work on how the method could be made more computationally efficient.
>
> Table 3: Testing-time robust generalization (RG %) on the last and best epochs, where DAC$_{t}$ indicates that decreasing adversarial certainty start from the $t$-th epoch.
> |RG (%)|Last|Best|
> |:-|:-:|:-:|
> |DAC$_{1}$|45.55|52.20|
> |DAC$_{101}$|45.84|50.49|
> |DAC$_{151}$|45.68|49.85|
>
> In the paper, we proposed DAC\_Reg to reduce the training cost, however, it modifies the optimization flow. Thus, to enhance the efficiency of DAC, we conduct an additional attempt by only decreasing adversarial certainty when robust overfitting happens. In our evaluation, we start the DAC from the 101-th and 151-th epochs, respectively, which correspond to the first and second times of learning rate decay. From Table 3, we can see that an earlier application of DAC brings better "Best" robustness, while the "Last" one is comparable. Consequently, improving training efficiency by reducing the epochs that consider DAC can only derive limited performance promotion. In other words, the potential direction of developing efficient methods might be regularizing some specific insights in a loss term, such as the design of DAC\_Reg, but the effort to guarantee stability is necessary.
>
> > [Potential limitation]
> > If possible, include detailed case studies or additional experiments where the method may not perform optimally. However, if resource constraints prevent additional experimentation, offering a discussion on potential limitations based on existing results would also help to clarify the boundaries of the method's effectiveness.
>
> Table 3 depicts that applying DAC earlier can achieve better robustness. Thus, DAC$\_{101}$ and DAC$\_{151}$, decreasing adversarial certainty when robust overfitting happens, cannot achieve comparable robustness on the best epoch with DAC$_{1}$ that works from scratch. That is, our method requires a warm-up, instead of an immediate functionality. Consequently, our method might not be suitable for the real-time scenario, where DAC is deployed only when robust overfitting happens, which limits the flexibility of our method.

---

> > ### Comment · Reviewer_XCB7 · 2024-09-30
> >
> > Thank you to the authors for their responses. They are reasonable, and I have no further questions. However, I do have one concern: the review of TMLR submissions indicates 'Requested Changes,' but the authors have not revised their manuscript. While this is acceptable to me, I am unsure if revisions are mandatory in this case.

---

> > > ### Comment · Reviewer_XCB7 · 2024-10-02
> > > **Updated after reading the uploaded revision**
> > >
> > > Thanks for the authors' revision. It is fine to me. I have no further comments.

---

> > > > ### Author Response · Authors · 2024-10-02
> > > > **Our Appreciation to Reviewer XCB7**
> > > >
> > > > We thank the reviewer's feedback and are happy that our response can solve the concerns of [Generalization to NLP], [Training efficiency], and [Potential limitation].

---

### Review · Reviewer_Nu3j · 2024-09-21

**Summary Of Contributions:**

This paper revisits the problem of robust overfitting in adversarial training and introduces the novel concept of adversarial certainty. The authors argue that model overconfidence in predicting adversarial examples during training is a key factor causing reduced robust generalization.

The paper contributes in:

Defining adversarial certainty to quantify the model’s certainty in adversarial predictions.

Proposing a method to generate less certain adversarial examples during training, which helps in improving robust generalization.

Providing theoretical results and empirical results that show decreasing adversarial certainty improves robustness across various image benchmark datasets.

**Audience:**

Yes

**Broader Impact Concerns:**

No.

**Claims And Evidence:**

Yes

**Requested Changes:**

Hyperparameter tuning on $\epsilon$. Please provide extra results on how the DAC method performs under different values of $\epsilon$. As $\epsilon$ directly affects the strength of the adversarial perturbations, understanding the model's performance across a range of $\epsilon$ values could give further insight into the robustness of your method under various adversarial conditions. After this, please also provide suggestions/discussions on $\epsilon$ choosing if the authors have.

Streaming issues:

In the contribution section, it is not recommended to cite equations or figures from later sections. Please try to summarize them instead. If you must cite them, you should provide details or define the variables in the equation before doing so.

Important notation, like adversarial certainty, is not properly defined until Chapter 4, yet it is frequently mentioned after just a one-line description. Please provide at least a high-level introduction before that.

Similarly, important methods like DAC_Reg are defined in Section 6.4, but are referenced earlier in Section 6.1.

**Strengths And Weaknesses:**

Strengths:

Innovative Concept: The introduction of adversarial certainty is novel and provides a way of understanding adversarial training dynamics.

Theoretical Results are Provided: The paper includes theoretical analysis to support the proposed approach, adding credibility.

Empirical Validation: Experiments across different datasets and adversarial training methods (AT, TRADES, MART) demonstrate the effectiveness of the method.

Weaknesses:

Robustness of $\epsilon$: One concern is how to choose $\epsilon$ prior to training. Do you have any suggestions on selecting $\epsilon$? Additionally, how does a change in $\epsilon$ affect performance?

The overall flow of the paper is not very strong. It consistently mentions concepts that have not yet been introduced, which leads to confusion.

---

> ### Author Response · Authors · 2024-09-30
> **Response to Reviewer Nu3j**
>
> We appreciate the reviewer's positive feedback on our innovative concept, theoretical results, and empirical validation. Below, we provide our responses to the required changes.
>
> > [Influence of $\epsilon$ on DAC]
> > Hyperparameter tuning on $\epsilon$. Please provide extra results on how the DAC method performs under different values of $\epsilon$. As $\epsilon$ directly affects the strength of the adversarial perturbations, understanding the model's performance across a range of $\epsilon$ values could give further insight into the robustness of your method under various adversarial conditions. After this, please also provide suggestions/discussions on $\epsilon$ choosing if the authors have.
>
> Table 1: Testing-time robust generalization (RG %) on the last and best epochs, where the training-time $\epsilon$ is set to $8/255$ while the testing one varies in $\\{4/255, 8/255, 12/255\\}$.
> |$\epsilon_\mathrm{tr}$=8/255|$\epsilon_\mathrm{te}$=4/255|$\epsilon_\mathrm{te}$=8/255|$\epsilon_\mathrm{te}$=12/255|
> |:-|:-:|:-:|:-:|
> |RG$\_\mathrm{last}$/RG$\_\mathrm{best}$|66.03/70.09|45.55/52.20|29.91/34.88|
>
> Table 2: Testing-time robust generalization (RG %) on the last and best epochs, where the testing-time $\epsilon$ is set to $8/255$ while the training one varies in $\\{4/255, 8/255, 12/255\\}$.
> |$\epsilon_\mathrm{te}$=8/255|$\epsilon_\mathrm{tr}$=4/255|$\epsilon_\mathrm{tr}$=8/255|$\epsilon_\mathrm{tr}$=12/255|
> |:-|:-:|:-:|:-:|
> |RG$\_\mathrm{last}$/RG$\_\mathrm{best}$|41.49/45.51|45.55/52.20|47.96/53.32|
>
> To investigate the influence of $\epsilon$ on our DAC method, we separately fixed the training- and testing-time $\epsilon$ to $8/255$, yet vary the other one in $\\{4/255, 8/255, 12/255\\}$, as shown in the above Tables 1 and 2. Specifically, Table 1 depicts that the model of a fixed training-time $\epsilon_\mathrm{tr}$ is more vulnerable to a larger testing-time $\epsilon$, on both the last and best epochs. On the other hand, Table 2 shows that a stronger adversarial ability in the training time can derives a better robustness when facing the same testing-time $\epsilon_\mathrm{te}$. Consequently, these results suggest that model robustness will benefit from a larger $\epsilon_\mathrm{tr}$ and a smaller $\epsilon_\mathrm{te}$. However, our paper mainly focuses on the $\epsilon=8/255$ in both the training and testing time for $\ell_{\infty}$-norm perturbations, since this setting is widely used in this field and $\epsilon$ is usually associated with the assumption of the adversarial strength.
>
> > [Streaming issues]
> > In the contribution section, it is not recommended to cite equations or figures from later sections. Please try to summarize them instead. If you must cite them, you should provide details or define the variables in the equation before doing so.
> > Important notation, like adversarial certainty, is not properly defined until Chapter 4, yet it is frequently mentioned after just a one-line description. Please provide at least a high-level introduction before that.
> > Similarly, important methods like DAC_Reg are defined in Section 6.4, but are referenced earlier in Section 6.1.
>
> We thank the reviewer for pointing out these streaming concerns. To improve the reading experience, we will revise our paper according to the reviewer's suggestions.

---

> > ### Comment · Reviewer_Nu3j · 2024-10-02
> > **Follow-up Response to Authors**
> >
> > Thanks for updating the revised paper. It is fine to me. I have no further questions.

---

> > > ### Author Response · Authors · 2024-10-02
> > > **Our Appreciation to Reviewer Nu3j**
> > >
> > > We thank the reviewer's feedback and are happy that our response can solve the concerns of [Influence of $\epsilon$ on DAC] and [Streaming issues].

---

### Author Response · Authors · 2024-10-01
**Update of Our Revised Paper According to Our Responses**

We hope our responses can solve the concerns proposed in the requested changes. Our paper has been revised according to our responses and the corresponding .pdf file has been updated, which aims to provide an easy way to check our changes.

---

### Decision · Action_Editor_pkex · 2024-10-14

**Recommendation:** Accept as is

**Comment:**

All the reviewers acknowledge the novelty of the adversarial certainty. The paper gives comprehensive theoretical analysis and empirical analysis to show the connection of adversarial certainty and robust generalization. All the reviewers acknowledge the proposed method to generate adversarial training examples with less certainty, and the correlation analysis  between adversarial certainty and robust
generalization. The experimental comparison is made on various datasets and strong baselines.

**Audience:**

Yes. The paper considers an interesting problem as adversarial training is effective to improve the robustness of deep learning models. The paper also proposes a new concept called adversarial certainty which is likely to draw attention from the community.

**Claims And Evidence:**

The paper proposes a novel concept called the adversarial certainty to capture the variance of the model’s predicted logits on adversarial examples. The paper shows that adversarial certainty is closely related to robust generalization, models with less adversarial certainty are more likely to generalize better. Based on this observation, the paper further proposes a method to decrease the adversarial certainty during the adversarial training, which can generate adversarial training examples with less certainty by solving a min-max-min problem. Extensive experimental analysis is conducted to show the efficiency of the proposed methods.